# Local topography increasingly influences the mass balance of a retreating cirque glacier

Caitlyn Florentine[1, 2], Joel Harper[2], Daniel Fagre[1], Johnnie Moore[2], Erich Peitzsch[1]

[1]U.S. Geological Survey, Northern Rocky Mountain Science Center, West Glacier, Montana, 59936, USA
[2]Department of Geosciences, University of Montana, Missoula, Montana, 59801, USA

*Correspondence to*: Caitlyn Florentine (caitlyn.florentine@umontana.edu)

**Abstract.** Local topographically driven processes, such as wind drifting, avalanching, and shading, are known to alter the relationship between the mass balance of small cirque glaciers and regional climate. Yet partitioning such local effects from regional climate influence has proven difficult, creating uncertainty in the climate representativeness of some glaciers. We address this problem for Sperry Glacier in Glacier National Park, USA using field-measured surface mass balance, geodetic constraints on mass balance, and regional climate data recorded at a network of meteorological and snow stations. Geodetically derived mass changes between 1950-1960, 1960-2005, and 2005-2014 document average mass change rates during each period at $-0.22\pm0.12$ m w.e. yr$^{-1}$, $-0.18\pm0.05$ m w.e. yr$^{-1}$, and $-0.10\pm0.03$ m w.e. yr$^{-1}$. A correlation of field-measured mass balance and regional climate variables closely (i.e. within 0.08 m w.e. yr$^{-1}$) predicts the geodetically measured mass loss from 2005-2014. However, this correlation overestimates glacier mass balance for 1950-1960 by $+1.18\pm0.92$ m w.e. yr$^{-1}$. Our analysis suggests that local effects, not represented in regional climate variables, have become a more dominant driver of the net mass balance as the glacier lost 0.50 km$^2$ and retreated further into its cirque.

## 1 Introduction

Glaciers are sensitive indicators of climate (Dyurgerov and Meier, 2000; Roe et al., 2016) because ice mass gains are ultimately controlled by winter precipitation, and ice mass losses are ultimately controlled by radiation and air temperature during summer. However, prior studies of small (i.e. <0.5 km$^2$) mountain glaciers, which are often located in cirques, show that local topographic effects, including avalanching, wind drifting, and shading, enhance winter mass gains or mediate summer mass losses (e.g. Hock, 2003; Kuhn, 1995; Laha et al., 2017). These topographically driven mass balance processes complicate the relationship between regional climate and cirque glacier surface mass balance.

Such complications have been documented in high relief areas worldwide. Analysis from Montasio Occidentale Glacier in the Italian Julian Alps demonstrated that avalanche-fed, shaded glaciers can exist at low elevations otherwise climatically unsuitable for the persistence of glacier ice (Carturan et al., 2013). Analysis from a network of on-ice automatic weather stations on a cirque glacier in the French Pyrenees concluded that topographic effects may exert more control on surface energy budgets - and thereby melt - than regional lapse rates in air temperature and moisture (Hannah et al., 2000). The small

glaciers of the North American Rocky Mountains have also been shown to have a disrupted, complex relationship to regional climate. In Colorado, glacier mass balance at very small (<0.2 km$^2$) glaciers showed no statistically significant correlation to winter precipitation during the 20$^{th}$ century, suggesting that winter mass inputs were not connected to regional winter precipitation in a straightforward, linear manner (Hoffman et al., 2007). This result instead implied the importance of
processes driven by local topography, including snow avalanching and wind drifting. In the Canadian Rockies, an inventory of nearly 2,000 glaciers showed that while larger glaciers retreated during the last half of the 20$^{th}$ century, very small glaciers did not change (DeBeer and Sharp, 2007). A follow-up analysis showed that the stability of these very small (<0.4 km$^2$) glaciers was closely related to their topographically favorable setting (DeBeer and Sharp, 2009).

Because local topography can substantially influence the mass balance of small and sheltered mountain glaciers, it follows
that as glaciers retreat further toward cirque headwalls, the direct control of regional climate on glacier mass balance should diminish, and local processes should become more influential (e.g. Haugen et al., 2010). That said, direct mass balance data from Andrews Glacier, an east-facing <0.2-km$^2$ glacier in Colorado, exhibited a strong correlation (r = -0.93) between summer temperature and net annual balance (Hoffman et al., 2007), despite enhanced accumulation via wind drifting and avalanching during the winter: glacier mass losses during summer outweighed the extra snow provided by topographically
driven snow redistribution. Mechanistically, this could be because Andrews Glacier is small enough to have a diminished katabatic boundary layer at its surface, leaving the glacier sensitive to ambient summer temperatures (Carturan et al., 2015). Regardless, this example shows that regional climate can remain the primary driver of net mass balance, even when local topography plays a strong role.

This paper examines the time evolution of the partitioning between regional climate and local influences on glacier mass
balance of a glacier retreating into its cirque. Since 1966, Sperry Glacier, Montana has reduced in area by 40% and retreated 300 m (Fagre et al., 2017). We hypothesize that during a recent interval (2005-2014), as a larger proportion of the glacier surface has become sheltered by its cirque headwall, the relationship between specific mass balance and regional climate is quantifiably distinct from that during mid-century (1950-1960), when the glacier extended further downslope from its cirque headwall. To test this hypothesis, we leverage a field measured (glaciological) surface mass balance record and repeat
geodetic mass balances.

## 2 Study area

Sperry Glacier (48.623° N, -113.758° W) sits just west of the Continental Divide, occupying a cirque that abuts the 2,801 m high Gunsight Peak, and is roughly in the center of Glacier National Park (GNP), Montana (Fig. 1a). A bedrock headwall, crested by a ridgeline that runs 2 km toward the northeast from Gunsight Peak (Fig. 1b), rises 100-300 m above Sperry
Glacier (Fig. 2). This headwall is 0.17 km$^2$ in area, and has slopes between 50° to near vertical. A sub-ridge extends 500 m toward the north, bordering the glacier's western margin (labelled west ridge in Fig. 1b). Between the headwall and this sub-

ridge, Sperry Glacier has a 40° ramp extending to the top of Gunsight Peak, so that some ice overlaps the ridge (Fig. 1b). A cornice that can be 10-20 m high develops every winter along this topmost section of the glacier (Fig. 2). A distinctive bergschrund separates the top 50-150 m of the glacier from the main ice body (Fig. 2). The terrain in front of Sperry Glacier is relatively low angle (<15°). Moraines located 1 km from the ice terminus (Fig. 2) indicate that in the geologically recent past, the glacier covered most of a topographic bench now bare of ice. These moraines were likely deposited when the glacier was at its Little Ice Age maximum extent (Carrara, 1989). Historical photographs and analysis from 1914 (Alden, 1914) show ice once covered 3.24 km$^2$. Since that time Sperry Glacier has steadily retreated, decreasing in area to 1.58 km$^2$ in 1938 (Johnson, 1980), and to 0.80 km$^2$ in 2015 (Fagre et al., 2017).

## 3 Methods

### 3.1 Elevation data

Modern Digital Elevation Models (DEMs) of Sperry Glacier and adjacent terrain were derived from National Technical Means imagery collected in 2005 and DigitalGlobe Worldview aerial imagery collected in 2014. These DEMs were generated by Fahey (2014), and are the same data as used by Clark et al. (2017). Both images were collected in September, when glacier mass is at an assumed annual minimum, on 02 September 2005 and 07 September 2014 respectively. The DEMs were generated with SOCET SET® software, which uses photogrammetric methods to extract terrain data from imagery (Zhang, 2006). Grid cell resolution was 5 m. The absolute accuracy of the DEM with respect to an independent vertical datum was 3.05 m in the horizontal and 7.54 m in the vertical. The precision of the DEM was 0.44 m and 0.47 m in the horizontal and vertical respectively.

U.S. Geological Survey (USGS) topographic maps at 6.1 m (20 ft) contour resolution were available for Sperry Glacier for 1950 and 1960 (Johnson, 1980). These maps were originally created using aerial photography and Kelsh plotter techniques, guided by 13 plane table bench marks. Both maps document elevation near the assumed September glacier mass minimum, on 01 September 1950 and 08 September 1960. We digitized these historic elevation data by first manually tracing scanned maps to produce digitized contours, and then interpolating digitized contours using a natural neighbor interpolation tool (ESRI, 2014; Sibson, 1981). Grid cell resolution was 5 m. We found that the historic elevation data from the original Johnson maps from 1950 and 1960 were 56 m lower than the modern vertical datum due to the authors' sea level datum assumption. We remedied this error by adding 56 m to the 1950 and 1960 DEMs.

### 3.2 Geodetic mass balance and DEM co-registration

The 1950 and 1960 DEMs did not require co-registration because they were originally mapped using a common vertical datum and spatial reference frame. We co-registered the 2005 and 2014, and the 1960 and 2005, DEMs following universal co-registration methods outlined by Nuth and Kääb (2011). The method used a statistical approach that minimized vertical differences over stable (i.e. not prone to landslide/rockfall), vegetation-free, snow-free, and low-sloping (< 30$^0$) bedrock

(Fig. S1, Supplement). After co-registration, the root mean squared error of elevation differences over stable bedrock improved from 8.43 m to 6.91 m for the 1960 and 2005 DEMs, and from 2.48 m to 2.01 m for the 2005 and 2014 DEMs. The mean of bedrock elevation differences after co-registration was effectively zero, i.e. $< 10^{-15}$ m, but differences over individual pixels still ranged from -42 m to +83 m for the 1960 and 2005 DEMs and -15 to +14 m for the 2005 and 2014 DEMs (Fig. S2, Supplement). These large differences tended to occur over steep terrain, and are included in our elevation error estimate.

Glacier margin data for 2005 and 2014, used to clip DEMs to glacier extent, were derived from Worldview aerial imagery (Fig. 1b) and ground-based GPS surveys of the glacier terminus. We defined 1950 and 1960 glacier margin data by digitally tracing the glacier from the same historic topographic maps by Johnson (1980).

We generated a record of geodetic mass balance for Sperry Glacier, given by:

$$B_a = \frac{\Delta V}{A}\left(\frac{\rho_i}{\rho_w}\right),\tag{1}$$

where $B_a$ is the specific surface mass balance expressed in meters of water equivalent, $\Delta V$ is the change in glacier volume (determined from DEM data described in Sect. 3.1), $\rho_i$ is the density of ice, $\rho_w$ is the density of water (1000 kg m$^{-3}$), and $A$ is the average of initial and final glacier area. To convert volume to mass, we adopted the approximation outlined by Huss (2013), and assigned ice density as $\rho_i = 850 \pm 60$ kg m$^{-3}$.

### 3.3 Geodetic mass balance assessment

There are three main sources of uncertainty in our geodetic mass balance calculation: elevation errors that affect the volume calculation, density errors which affect the volume to density conversion, and map coverage errors that affect the historic geodetic mass balance.

### 3.3.1 Elevation change uncertainty

After co-registering DEMs, elevation error on the geodetic mass balance was estimated by analyzing elevation differences over stable bedrock terrain. Because error tends to be greater over steep terrain at high elevations, we analyzed error by 50-m elevation band, rather than in bulk across the entire DEM. The quantity of stable bedrock points varied from 14,472 in the lowest elevation band to 1,825 points in the highest elevation band. We used standard error propagation, as applied in previous geodetic mass balance studies (Ruiz et al., 2017; Thomson et al., 2017), given by:

$$E\Delta h_i = \frac{\sigma_{dh}}{\sqrt{N_{eff}}},\tag{2}$$

where $E\Delta h_i$ is the mean elevation error for a 50-m elevation band ($i$) that spans the glacier, $\sigma_{dh}$ is the standard deviation of elevation differences over stable bedrock for the elevation band, and $N_{eff}$ is the number of independent values within the elevation band. This in turn is given by:

$$N_{eff} = \frac{N_{tot} \cdot P}{2 \cdot d} \tag{3}$$

where $N_{tot}$ is the total number of pixels (grid cells), $P$ is pixel size, and $d$ is the distance of spatial autocorrelation (100 m), estimated from variogram analysis (Smith et al., 2018). We then summed elevation error across the glacier surface, weighting the error by the ratio of the glacier surface covering that elevation band.

### 3.3.2 Density uncertainty

We followed the convention outlined by Huss (2013) and used a constant density of $\rho_i$= 850±60 kg m$^{-3}$ to convert glacier volume change to mass, which was derived from a suite of empirical firn densification experiments, and is appropriate for geodetic mass balance calculations where the time intervals considered are longer than 5 years, the volume change is nonzero, and the mass balance gradient is stable. These conditions are satisfied for Sperry Glacier because (a) geodetic mass balance periods are 10 years (1950-1960), 45 years (1960-2005), and 9 years (2005-2014) long, (b) glacier volume changes were nonzero, and (c) glaciological measurements show that mass balance gradients were relatively constant during 2005-2014, i.e. 0.004-0.019 m w.e. m$^{-1}$ for winter and 0.003-0.011 m w.e. m$^{-1}$ for summer (Clark et al., 2017). The ±60 kg m$^{-3}$ density error amounted to < 8% error on geodetic mass balances.

We assumed that elevation and density errors were independent, and therefore summed them quadratically to solve for the total error on the geodetic mass balance, given by:

$$E_t = \sqrt{E\Delta h^2 + E\rho^2} \tag{4}$$

where $\boldsymbol{E\Delta h}$ is elevation error, $\boldsymbol{E\rho}$ is density error, and $\boldsymbol{E_t}$ is the total error in m w.e. on the geodetic mass balance.

### 3.3.3 Map coverage errors

The historic maps used to derive 1950 and 1960 DEMs are missing the upper section of the glacier. Historic photos verify that in the mid-20[th] century Sperry Glacier extended to the top of Gunsight Peak, as it does today. To enable consistent geodetic mass balance calculation for the entirety of Sperry Glacier, we filled in elevation change over this missing section (Fig. 3) using modern 2005-2014 results. We opted for this remedy, rather than the alternative of truncating 2005-2014 data, in order to be consistent with glaciological data, which were generated for the entire glacier surface including the upper section. By using modern elevation change rate results to infill missing historic data, we assumed that the rate of mass change in that area near the cirque headwall was the same through the study interval (1950-2014). Given that we had no way

to test the validity of this assumption, and to ensure it did not fundamentally alter geodetic mass balance results, we also computed results for the truncated glacier (Table S1, Table S2, Supplement). The difference between the truncated and infilled geodetic mass balance was ≤0.04 m w.e. yr$^{-1}$ for both 1950-1960 and 1960-2005 (Table S2, Supplement), which is less than the accounted geodetic mass balance error (0.05 m w.e. yr$^{-1}$ for 1950-1960, 0.12 m w.e. yr$^{-1}$ for 1960-2005) (see section 4.1). Infilling data for the missing upper section therefore does not alter geodetic mass balance results beyond the reported uncertainty bounds.

### 3.4 Glaciological mass balance

Sperry Glacier has been the focus of a USGS intensive glacier mass balance monitoring program since 2005. Therefore, glaciological mass balance measurements of seasonal mass gains and losses at Sperry Glacier are available from the year 2005 onward (Clark et al., 2017). We used these data, measured in the field according to standard mass balance protocols (Kaser et al., 2003; Ostrem and Brugman, 1991) to define specific, conventional (Cogley et al., 2011) winter and summer glacier mass balance from 2005-2014. We also analyzed point balance data collected along a longitudinal transect (stakes in Figure 1; data from Clark et al. 2017) to inspect mass balance gradients at Sperry Glacier. To correct for bias in the raw, specific glaciological balances reported by Clark et al. (2017), we performed calibration as outlined by Zemp et al. (2013). The results of this calibration, which utilized the 2005-2014 geodetic mass balance calculated in this study to correct the absolute magnitude of annual and summer glaciological balances without losing seasonal/annual variability (Zemp et al., 2013), are reported in Table 1. Such calibration ensures that systematic errors in the glaciological method are rectified and englacial and subglacial mass changes not measured at surface stakes are accounted. Details on this calibration are provided in the Supplement.

### 3.5 Mass balance and climate regressions

Employing annual glaciological measurements (Clark et al., 2017), we defined a functional relationship (i.e. linear regression) between 2005-2014 specific surface mass balance and regional climate. Regional climate was defined by data on winter peak snow water equivalent (SWE) and summer positive degree days (PDD). Climate data from 2005-2014 were used to define the regressions, and climate data from 1950-2005 were used to apply the regressions back in time.

### 3.5.1 Meteorological and snow data

The Global Historical Climatology Network provides historic temperature data for Kalispell, Montana located approximately 50 km southwest of Sperry Glacier (Figure 1). There are closer meteorological stations, but these had short and incomplete records. For example, average daily temperatures were recorded at Sperry Glacier for most (83%) summer days from 2005-2013 (Baker et al., 2018). To assess the representativeness of Kalispell temperature data relative to the glacier site, we compared this shorter, discontinuous record with the longer, continuous Kalispell record. Temperatures measured at Kalispell are highly representative of regional climate as reflected by gridded values from North American Regional

Reanalysis output (Supplement, Fig. S3a). Kalispell is located at an elevation of 901 m, which is 1554 m lower than the average elevation across Sperry Glacier in 2005-2014. We therefore applied a lapse rate correction of -5.57 °C km$^{-1}$, calculated using 7 years of temperature measurements from a meteorological station at Sperry Glacier (Figure 1b), to correct for the elevation difference. Comparison of lapse-rate-corrected temperatures derived from the Kalispell meteorological station with the 7-year record of in situ July, August, September temperatures measured at the Sperry Glacier meteorological station (Fig. 1b) yielded no statistically significant difference at the 99% confidence interval (p<0.01), and the distribution of residuals was normal.

The Kalispell record reports continuous daily average air temperatures back to 1950 for the melt season summer months. We follow the convention defined by previous GNP melt modelling (Clark et al., 2017) and confine the Sperry Glacier melt season to July, August, and September. Glaciological observations suggest that although May and June can be warm enough to generate melt at Sperry Glacier, the deep snowpack is not necessarily warmed with pore space filled to saturation, and therefore melt does not necessarily run off the glacier.

Because the glacier terminus increased in elevation from 1950-2014, the surface of Sperry Glacier was on average 35 m lower in elevation during 1950-1960, and 21 m lower in elevation during 1960-2005. We accounted for these changes in elevation by adjusting the lapse rate correction accordingly for each time period. The effect of this accounting was small, resulting in -0.16°C and -0.12°C changes to average summer temperature, and cumulatively -24 °C day and -14 °C day changes to PDD. Ultimately, correcting for elevation changes at Sperry Glacier corresponded to cm-scale changes to the mass balance regression.

Data from the Natural Resources Conservation Service at five nearby snow course sites (Desert Mountain, 1707 m; Piegan Pass, 1676 m; Marias Pass, 1600 m; Mount Allen, 1737 m; and Mineral Creek, 1219 m) and one adjacent Snow Telemetry (SNOTEL) site (Flattop Mountain, 1921 m) provide SWE data for locations within a 50-km radius of Sperry Glacier (Fig.1a). Temperature data are also recorded at the Flattop Mountain site. SWE is recorded monthly at snow course sites, via manual measurement, whereas SWE is recorded daily at SNOTEL sites via transducer measurements from a snow pillow. The 1950-2014 period is long enough to encompass meso-scale changes in atmospheric flow patterns, which elsewhere have been shown to have an important impact on winter accumulation and glacier mass balance (e.g. Huss et al., 2010). The peak SWE data we analyzed have been shown to reflect such meso-scale, decadal shifts in snow (McCabe and Dettinger, 2002; Pederson et al., 2011; Selkowitz et al., 2002).

Based on an analysis of the historical snow data from the region against observations from Sperry Glacier, we chose to use Mount Allen snow data for our mass balance regression. We found high correlation coefficients (r$^2 \geq$ 0.939) for each of the seven datasets analyzed (Table S3, Supplement). The highest correlation was with Flattop Mountain SNOTEL (r$^2$ = 0.990),

but at this station consistent snow data only go back to 1970. The second highest correlation ($r^2 = 0.973$), for a snow record that started prior to 1950 and so encompassed the geodetic record, was Mount Allen.

### 3.5.2 Regression analysis

To quantify the glacier-climate relationship for 2005-2014, we fit a linear regression between 2005-2014 climate data (x: PDD, SWE) and glaciological mass balance (y: $b_s$, $b_w$) data. We forced the best-fit line through the origin so that zero PDD and zero SWE equated to zero melt and zero accumulation, respectively. The linear combination of the two seasonal linear equations was thus:

$$B_a = m_s (PDD) + m_w (SWE) \tag{5}$$

where $B_a$ is the specific annual balance, $m_s$ is the summer proportionality factor, $m_w$ is the winter proportionality factor, PDD is summer positive degree days, i.e. the sum of the average temperatures of days above 0 °C during the melt season, and SWE is peak winter snow water equivalent. We solved for the proportionality factors using an ordinary least squares, one parameter linear regression.

The linear regression only provides a best estimate of the mass balance of Sperry Glacier. To discern how dependable this estimate was, we considered the 95% confidence interval on each seasonal linear regression. Knowing that the true proportionality factor (i.e. slope of the best fit line) fell somewhere within the upper and lower bounds, we used the upper and lower confidence intervals to compute maximum and minimum possible mass balances. This accounting accommodated uncertainty due to (a) the discrepancy between snow and meteorological station locations and Sperry Glacier, (b) assuming that seasonal melt is largely driven by net available shortwave radiation and that PDD is a reasonable proxy for this, and (c) assuming that the summer melt season is limited to July, August, and September (JAS). To test the sensitivity of our results to this last assumption, we computed a summer linear regression using May, June, July, August, and September (MJJAS) PDD. Ultimately, the median difference between mass balance values produced by the MJJAS versus JAS regressions was just 0.04 m w.e. yr$^{-1}$.

The linear regression quantifies the 2005-2014 relationship between Sperry Glacier and regional climate, and is fixed with respect to this time. Yet we have hypothesized that this relationship changed as the glacier retreated. By fixing the proportionality factors $m_s$ and $m_w$ to the modern glacier-climate relationship, and then forcing this modern regression with historic climate data, we test our hypothesis that the glacier-climate relationship changed as the glacier retreated.

### 3.6 Shading, avalanching, and wind-drifting

Attributing the hypothesized change in the glacier-climate relationship to the increasing influence of local topographic effects requires inspection of topographically influenced processes at Sperry Glacier. To assess topographic shading, we used the Solar Area Radiation tool (ESRI, 2014) to calculate cumulative insolation received across Sperry Glacier during the melt season months of July, August, and September in 2014, when the glacier was smaller, steeper, and more shaded, versus 1950, when the glacier was larger, flatter, and less shaded. This hemispherical viewshed algorithm (Fu and Rich, 2002) calculated radiation based on an upward-looking sky map for every grid cell within our 2014 DEM, the seasonal progression of the position of the sun relative to Earth, a fixed atmospheric transmissivity, topographic shading, latitude, elevation, slope, and aspect. To qualitatively assess avalanching and wind-drifting snow processes, we examined field observations, historic photographs, and field-measured mass balance data collected by Clark et al. (2017).

## 4 Results

### 4.1 Retreat, thinning, and negative mass balance

Sperry Glacier has decreased from 1.30 km$^2$ in 1950, to 1.23 km$^2$ in 1960, to 0.86 km$^2$ in 2005, to 0.80 km$^2$ in 2014 (Table 2). During the mid-20$^{th}$ century the glacier extended onto relatively flat (<15° slope) bedrock terrain, therefore the lower portion of the glacier was relatively low angle. As a result, between 1950 and 2014, despite nearly 0.5 km of retreat, the glacier terminus only receded upward by 56 m in elevation. The loss of this northwest-oriented, low-sloping terminus resulted in a steepening of the glacier's median slope by 9$^0$, and a rotation of the glacier's median aspect toward the north by 11$^0$.

Elevation change across the glacier surface is generally similar in its spatial pattern, but not magnitude, during 1950-1960, 1960-2005, and 2005-2014 (Fig. 3a, c, e). Thinning occurred across the lower portion, but is most pronounced at the terminus. The upper elevations of the glacier thickened from 1960-2005 and 2005-2014, but at rates less than +1 m yr$^{-1}$ (Fig. 3). The magnitude of thinning near the terminus is distinct for each period. Terminus thinning rates from 1950-1960 were up to -2.5 m yr$^{-1}$ (Fig. 3b), whereas terminus thinning rates from 1960-2005 (Fig. 3d) and 2005-2014 (Fig. 3f) were smaller than -1.5 m yr$^{-1}$. No development of debris cover at the glacier terminus that might explain this decreased thinning rate is evident. The magnitude of thickening was likewise distinct between periods, with 1950-1960 showing a bulge near the glacier's middle (approximately 2500 m), growing at nearly +1 m yr$^{-1}$ (Fig. 3b).

Despite differences in the magnitude of elevation change, the hypsometry of the glacier remained similar in 1950, 1960, 2005, and 2014. Sperry Glacier lost 50 m of ice at the glacier terminus during the 1960-2005 interval which agrees with the average amount of thinning (52.4 m) reported for glaciers in the Canadian Rockies for the same period (Clarke et al., 2017). Commensurate with its area loss and thinning, the glacier also lost volume. Geodetic mass balance results show Sperry Glacier shrank by -3.33 × 10$^6$ m$^3$ from 1950-1960, -11.3 × 10$^6$ m$^3$ from 1960-2005, and -0.90 × 10$^6$ m$^3$ from 2005-2014

(Table 3). It lost $-2.83 \times 10^9$ kg of mass from 1950-1960, $-9.68 \times 10^9$ kg from 1960-2005, and $-0.76 \times 10^9$ kg from 2005-2014 (Table 1). The rate of mass change at Sperry Glacier was $-0.22 \pm 0.12$ m w.e. yr$^{-1}$ from 1950-1960, $-0.18 \pm 0.05$ m w.e. yr$^{-1}$ from 1960-2005, and $-0.10 \pm 0.03$ m w.e. yr$^{-1}$ from 2005-2014 (Table 3). The glacier is near balance, but slightly losing mass.

<p>5</p>

## 4.2 Glacier-climate relationship

Linear regressions show strong ($r^2 > 0.97$), statistically significant ($p < 0.03$) correlation between climate and glaciological data (Fig. 4). The model therefore effectively defines a functional relationship between glacier mass balance and regional climate for 2005-2014. The regressions correlate warmer summers to more negative summer mass balance ($r^2 = 0.978$), and snowier winters to more positive winter mass balance ($r^2 = 0.973$). From the regression results, the proportionality factor for winter ($m_w$), which scales snow course data on peak SWE to winter glaciological measurements, is 2.99. The proportionality factor for summer ($m_s$), which scales meteorological data on PDD to summer glaciological measurements, is -0.004 m w.e. °C$^{-1} \cdot$ day$^{-1}$.

We used climate data to apply the linear regression back in time to 1950 (Figure 5). Average PDD during 1950-1960 was $709 \pm 53$ °C $\cdot$day, which was virtually the same as average PDD during 1960-2005 ($717 \pm 86$ °C $\cdot$ day). Summer PDD then showed a 41 °C $\cdot$ day increase to an average of $758 \pm 75$ °C $\cdot$ day during 2005-2014, although this increase was well within interannual variability. Snow data show that on average, SWE decreased from $1.31 \pm 0.26$ m w.e. in 1950-1960, to $1.08 \pm 0.32$ m w.e. in 1960-2005, to $0.95 \pm 0.25$ m w.e. in 2005-2014, although these step decreases are likewise within the range of interannual variability. The step change in peak SWE during the mid-1970s is consistent with other regional SWE records (Fig. S3b, Supplement), and has been interpreted as a result of a modal change in the Pacific decadal oscillation, a pattern of ocean climate variability which is closely tied to peak SWE in this region (McCabe and Dettinger, 2002; Pederson et al., 2011; Selkowitz et al., 2002). Generally, PDD and SWE in 1950-1960 compared to 2005-2014 seem to differ. To quantitatively assess the difference, we performed simple t-tests. T-test results showed that the lower (higher) average SWE (PDD) during 2005-2014 differed statistically from 1950-1960 SWE (PDD) at $p < 0.01$, $\alpha = 0.99$ ($p < 0.15$, $\alpha = 0.85$). This evidence supports the notion that 2005-2014 had relatively dry winters and warm summers compared to 1950-1960.

Nevertheless, geodetic results show that average mass change rates at Sperry Glacier for 2005-2014 ($-0.10 \pm 0.03$ m w.e. yr$^{-1}$) and 1950-1960 ($-0.22 \pm 0.12$ m w.e. yr$^{-1}$) were comparable, i.e. within error bounds. This differs drastically from mass balance results derived from the regression, which include many years of net mass gain during the mid-20th century (Fig. 5). Sufficient individual years are positive during the mid-20th century to yield positive averages from 1950-1960 and 1960-2005, at $+0.98 \pm 0.83$ m w.e. yr$^{-1}$ and $+0.34 \pm 0.75$ m w.e. yr$^{-1}$ respectively. However, confidence intervals from the linear regression model suggest the possibility of a negative average during 1960-2005. Conversely, error on the 1950-1960 mass balance include only positive averages, i.e. glacier thickening and mass gain. The 1950-1960 mass gain predicted by the

regression (+0.98 ± 0.83 m w.e. yr$^{-1}$) is distinctly at odds with the 1950-1960 geodetic mass balance of -0.22 ± 0.12 m w.e. yr$^{-1}$.

## 4.3 Local controls on surface mass balance

The amount of potential clear sky radiation available for specific summer melt at Sperry Glacier decreased by 118,605 kJ m$^{-2}$ (approximately 15 W m$^{-2}$) from 1950-2014, likely due to steepening of the glacier surface, and a greater proportion of the glacier becoming shaded. Given the heat of fusion for ice (334 kJ m$^{-2}$), this energy deficit translates to 0.36 m w.e. less potential melt for the summer melt season averaged over the entire glacier, driven only by changes in the relative influence of local effects (i.e. shading, steepening), independent of climate. Field observations show that the rock headwall extending above Sperry Glacier (Fig. 2) contributes high frequency, low volume, loose avalanches (Fig. 6b), and that the cornice above the bergschrund often breaks in the spring, subsequently triggering localized slab avalanches that extend 500 m down from the crest of the ridge to an elevation of 2460 m. (Fig. 6c). Historic aerial photographs show that a prominent ridge of wind-drifted snow consistently develops at lower elevations in the basin (Fig. 6a). Field-measured mass balance data showing highly variable snow accumulation provide evidence of wind effects. Accumulation ranges from 0.00 m w.e. in wind scoured areas, where seasonal snowpack had been stripped down to bare ice, up to more than 5 m w.e. in wind loaded areas (Clark et al., 2017).

Field-measured point data (n=551), taken along stakes from the terminus toward the headwall of the glacier (Clark et al., 2017), show the impact of these local effects (shading, avalanching, wind drifting) on the mass balance elevation gradient (Fig. 7c). Toward the glacier terminus, at elevations lower than 2475 m, the mass balance gradient of winter accumulation is $10 \times 10^{-4}$ m w.e. (m)$^{-1}$. Toward the glacier head, at elevations above 2475 m, the mass balance gradient of winter accumulation is an order of magnitude higher at $150 \times 10^{-4}$ m w.e. (m)$^{-1}$. The elevation gradients shown by summer ablation measurements show a similar inflection (Fig. 7c). At elevations higher than 2475 m, where the glacier is steeper and more shaded, the melt gradient rate increases eightfold from $6.6 \times 10^{-4}$ m w.e. (m)$^{-1}$ to $53 \times 10^{-4}$ m w.e. (m)$^{-1}$.

## 5 Discussion

Our results reveal that the drivers of mass balance at Sperry Glacier evolved as the glacier retreated. Specifically, the strong correlation between modern-era field and climate data poorly predicts the 1950-1960 geodetically derived mass balance (Fig. 5c) and documents a change in the relationship between regional climate and rates of ice mass loss.

It seems that Sperry Glacier became less reflective of the regional climate as it retreated. Geodetic results compared to regional climate data show that Sperry Glacier lost less area-averaged mass from 2005-2014 (-0.10 ± 0.03 m w.e. yr$^{-1}$) than from 1950-1960 (-0.22 ± 0.12 m w.e. yr$^{-1}$) despite the 2005-2014 period being characterized by warmer summers and lower precipitation winters. Glacier elevation changes reflect both surface mass balance and ice flow processes (Cuffey and

Patterson, 2010). However, if we were to attribute geodetic results solely to surface mass balance, then our results (Fig. 3) suggest that the equilibrium line altitude remained relatively constant from 1950-2014 despite climate warming. Thus as Sperry Glacier retreated, its accumulation area ratio (Cogley et al., 2011) increased. With an increased fraction of the glacier remaining snow-covered throughout the melt season, the average glacier albedo increases (Naegeli and Huss, 2017). Such time changes in glacier albedo must have affected the summer proportionality factor, i.e. the amount of area averaged melt relative to regional summer temperature, which in part explains the discrepancy between the linear regression and the 1950-1960 mass balance (Fig.5c).

We had hypothesized that the discrepancy between the regression and the historic mass balance would be attributable to avalanching, wind drifting, and shading becoming relatively more influential as the glacier retreated. Glaciological measurements provide direct evidence of these effects, as the mass balance gradient at Sperry Glacier has two distinct components reflecting regional and local drivers (Fig. 7c). For example, the winter mass balance gradient at elevations below 2475 m, $10 \times 10^{-4}$ m w.e. $(m)^{-1}$, falls within the range of regional SWE lapse rates, $6.2\text{-}10.4 \times 10^{-4}$ m w.e. $(m)^{-1}$, reported in a study of snow accumulation in northwest Montana (Gillan et al., 2010). The stark increase (to $150 \times 10^{-4}$ m w.e. $(m)^{-1}$) in winter mass balance gradient at higher elevations is consistent with enhanced snow accumulation reported for so-called "drift" glaciers located below the regional equilibrium line altitude in Colorado, where winter accumulation from local effects was four to eight times regional snow accumulation (Outcalt and MacPhail, 1965). Field-measured summer data at high elevations are sparse, but these summer point balances also show a sudden steepening of the summer mass balance gradient above 2475 m (Fig. 7c), likely because these high reaches of the glacier are shaded by the headwall.

Our linear regression results fall short of elucidating what drove the change in the relationship between climate variables and Sperry Glacier. Therefore, we used glaciological measurements in one final, complementary analysis to assess the impact of the time-changing glacier hypsometry and the local and regional mass balance gradients on the glacier's total mass balance. We applied field-measured mass balance gradients to the hypsometry of Sperry Glacier in 2014 and 1950 (Fig. 7a,b black bars), first using only the regional gradient (Fig. 7c dotted lines) and then using the regional plus local gradients (Fig. 7c solid lines). This demonstrates that without local effects, given the 1950 glacier hypsometry, specific surface mass balance would be -1.04 m w.e. Similarly, without local effects, given the 2014 glacier hypsometry, specific surface mass balance would be -0.96 m w.e. Thus, in theory, the hypothetical mass balance response of the 1950 and 2014 glaciers would be roughly similar when forced only by regionally determined gradients. However, when the local mass balance gradients are applied, the balance for 1950 increased by 37% to -0.66 m w.e., and for 2014 by 57% to -0.41 m w.e. These results demonstrate that local mass balance processes have apparently played a strong role at Sperry Glacier since 1950, and that role strengthened as the glacier retreated.

The impact of topographic effects is not evenly partitioned between seasonal components. The differences between the solid and dotted lines in Figure 7b and Figure 7d illustrate local effects: suppression of summer melt (light red area), surplus in

winter accumulation (light blue area), and net mass balance enhancement (light gray area). We find that winter local effects (i.e. the surplus in winter accumulation due to avalanching and wind loading) account for 79% of the discrepancy between the mass balance defined by regional mass balance gradients alone versus that defined by both regional and local gradients. Summer local effects (i.e. the mediation in summer melt due to shading) accounted for 21% of the discrepancy. This analysis

supports the interpretation that the altered mass balance response we have documented, wherein the glacier had a less negative balance in 2005-2014 despite less favorable regional climate conditions, is driven by the increasing influence of local effects rather than just the changing glacier hypsometry. Here we have examined the time evolution of local effects in bulk. To quantitatively partition the mass impact of discrete processes, future work could assess the evolution in geographic parameters for wind (e.g. Winstral et al., 2002) and avalanche (e.g. Carturan et al., 2013) effects.

Sperry Glacier's increasing sensitivity to local mass balance drivers is consistent with studies of 20[th] century glacier change elsewhere in the Rocky Mountains (DeBeer and Sharp, 2009), and is commensurate with modeled projections of future changes to cirque glaciers in the Swiss Alps. For example, although 25% of 1133 individual very small (<0.5 km$^2$) glaciers are projected to disappear as soon as the next 25 years, 67 glaciers (6%) are projected to maintain more than half of their area through at least 2050 (Huss and Fischer, 2016). The persistence of these select glaciers, which are located at lower elevation

than the regional equilibrium line altitude, signals the sometimes strong influence of local mass balance drivers. Despite local effects allowing some glaciers to persist for a few more decades, categorical evidence of world-wide glacier retreat in response to 20[th] century warming (Roe et al., 2016) suggests that local mass balance drivers do not interrupt the synchronicity of glacier response to climate change on global, century-long scales. Indeed, very small (<0.2 km$^2$) glaciers in Colorado showed a strong annual mass balance response to 20[th] century summer temperatures despite being heavily

influenced by winter topographic effects (Hoffman et al., 2007). Thus, the evolving relationship between climate and mass balance demonstrated by Sperry Glacier reveals the complexity in interpreting glacier changes in Glacier National Park and at small cirque glaciers elsewhere on Earth, but does not preclude the reality of a climate that is trending toward conditions that mandate glacier disappearance.

**6 Conclusion**

Analysis of a 64-year record of glacier mass change against meteorological and snow data demonstrates that the relationship between regional climate variables and Sperry Glacier is evolving through time. By assessing geodetic mass measurements, regional climate data, and field measured mass balance, we deduced that this shift was caused by local drivers related to topography becoming increasingly influential as the cirque glacier retreated. Our results therefore emphasize the importance of accounting for spatially complex, local topographic processes in projections of 21[st] century mountain glacier change.

These effects can exert substantial and time-changing control on the mass balance of retreating cirque glaciers, are likely highly variable from glacier to glacier, and must therefore be carefully considered and treated in interpretations and projections of cirque glacier change.

**7 Disclaimer**

Any use of trade, firm, or product names is for descriptive purposes only and does not imply endorsement by the U.S. Government.

*Data Availability.* Temperature data from the Kalispell Historical Climate Network site are available at
https://www.ncdc.noaa.gov/data-access/land-based-station-data/land-based-datasets/us-historical-climatology-network-ushcn. Temperature data from the Sperry Glacier meteorological station are available at https://doi.org/10.5066/F7BG2N8R in version 2.1. Snow data analyzed in this study are available at https://www.wcc.nrcs.usda.gov/snow/, which also has the temperature data for the SNOTEL site at Flattop Mountain. North American Regional Reanalysis data analyzed in this study were provided by the NOAA/OAR/ESRL PSD, Boulder, Colorado, USA from their web site at
https://www.esrl.noaa.gov/psd/.

**The Supplement related to this article is available online at doi: xxx-supplement.**

*Competing interests.* The authors declare that they have no conflict of interest.

*Author contributions.* CF and JH designed the analytical approach, which CF then carried out with input from DF, JM, and EP. CF prepared the manuscript with contributions from all co-authors.

*Acknowledgements.* This work was funded by the Climate and Land Use Change mission area of the U.S. Geological Survey. CF received funding from the Jerry O'Neal National Park Service Student Fellowship.

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

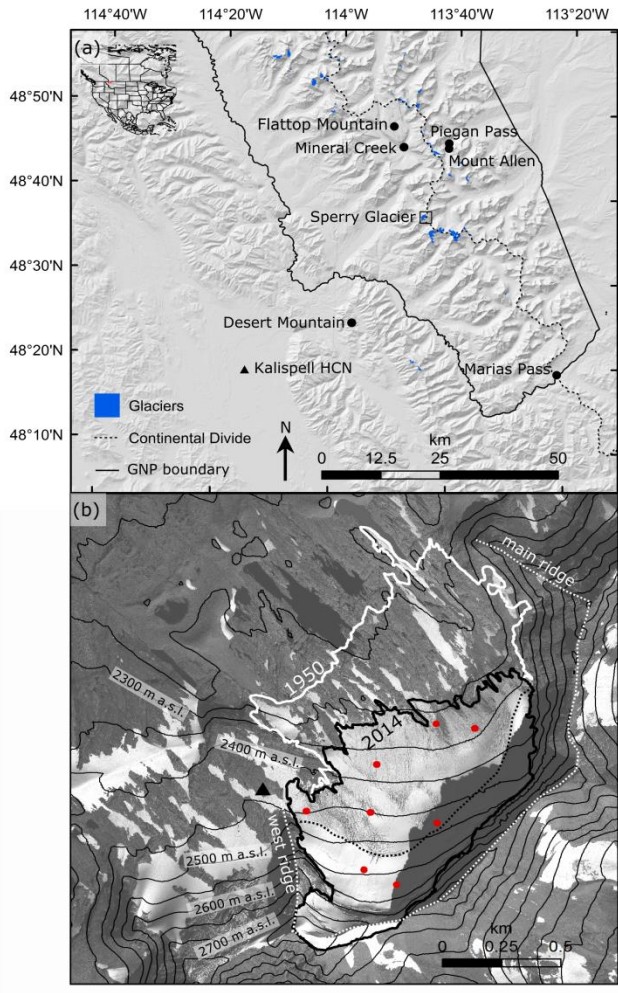

**Figure 1:** (a) Study area in the U.S. Northern Rocky Mountains, located just south of the USA-Canada border, as indicated by the red box on the inset map. The 39 named glaciers within and near Glacier National Park (GNP) are shown (blue). The continental divide (dashed black line), GNP boundaries (solid black line), the location of Sperry Glacier (small box), Kalispell Historical Climate Network (HCN) meteorological station (black triangle), and snow measurement sites (black circles), are also shown. (b) Worldview aerial imagery from 07 September 2014 depicts Sperry Glacier with the 1950 (white line) and 2014 (black line) glacier margins shown. The approximate seasonal snow line (dotted black line) and the main ridge that defines the crest of the cirque headwall above Sperry Glacier, as well as the west ridge which bounds Sperry Glacier (dotted white line), are also depicted. Elevation (m) above sea level (a.s.l.) is represented by thin black contour lines. Stakes where glaciological measurements were made are shown (red) as is the Sperry Glacier meteorological station (black triangle).

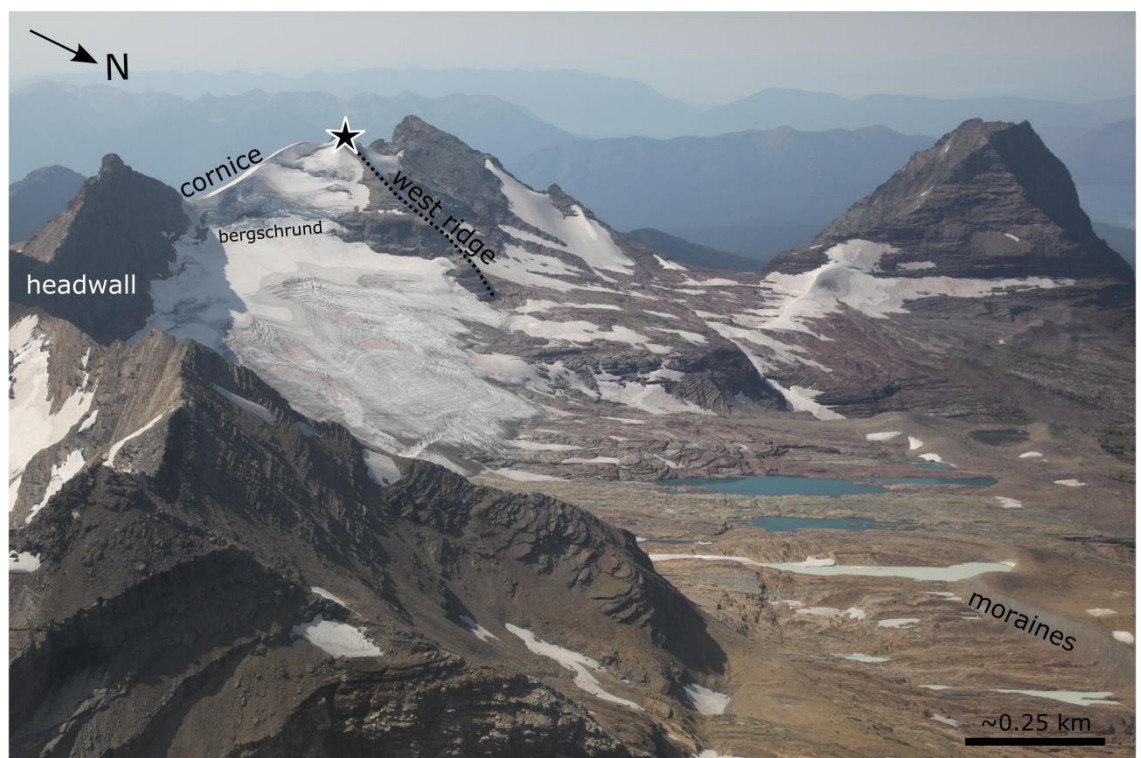

**Figure 2:** Sperry Glacier on 01 September 2009. Arrow points to the north. Note low-sloping proglacial terrain. In 1950 and 1960, the glacier extended onto this relatively flat ground. Gunsight Peak is indicated by the black star. The west ridge, headwall, cornice, bergschrund, and moraines discussed in the text are labeled. Photograph credit: USGS Climate Change in Mountain Ecosystem photograph archives.

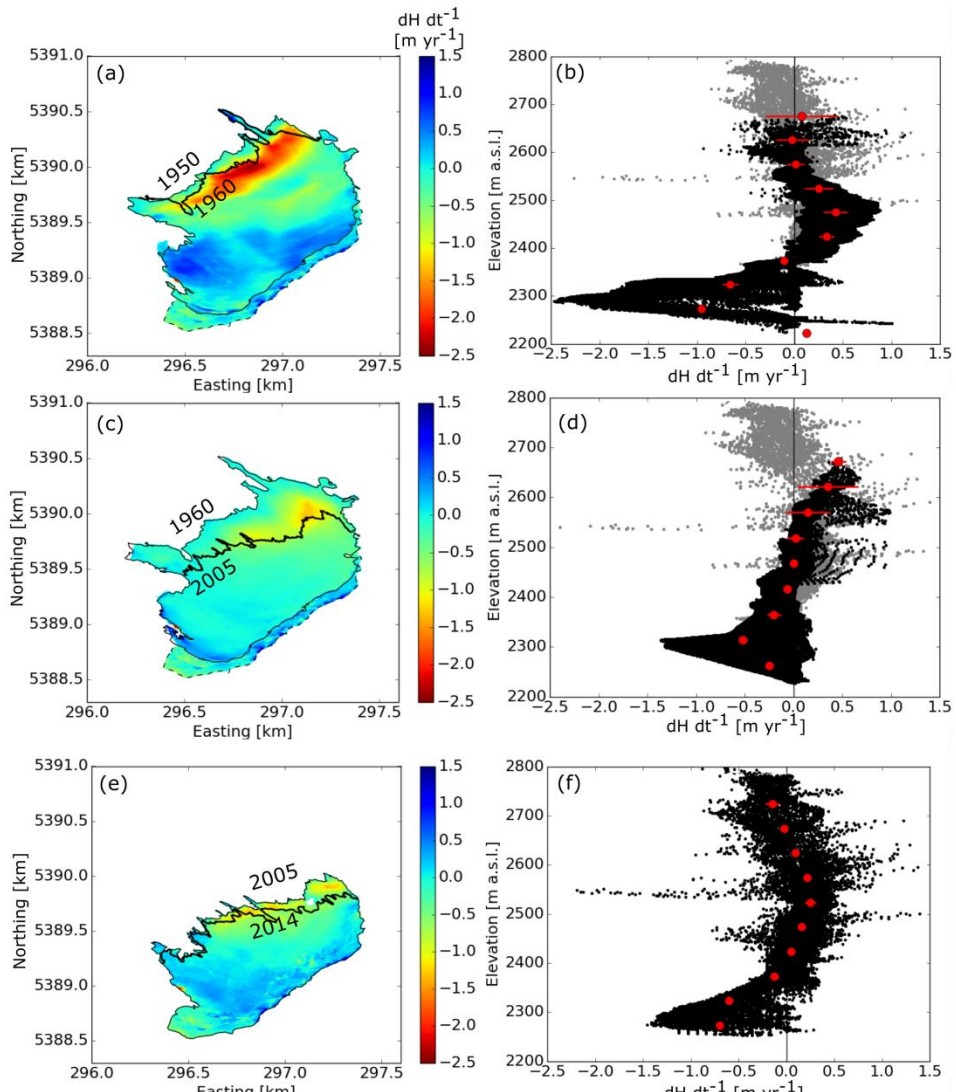

**Figure 3:** Elevation change over Sperry Glacier from (a,b) 1950-1960, (c,d) 1960-2005, and (e,f) 2005-2014. Left column shows elevation change across the glacier surface, and right column shows hypsometry of elevation change. Black dots on the hypsometry figures (b,d,f) indicate individual pixels, red dots indicate elevation band means, and horizontal red bars indicate elevation band mean errors. Error is mostly smaller than, and therefore obscured by, the red dot. The missing upper section is delineated by the dotted line in (a) and (c). Data from 2005-2014 were used to fill in this missing section. These 2005-2014 infill data are indicated by the gray dots in (b) and (d).

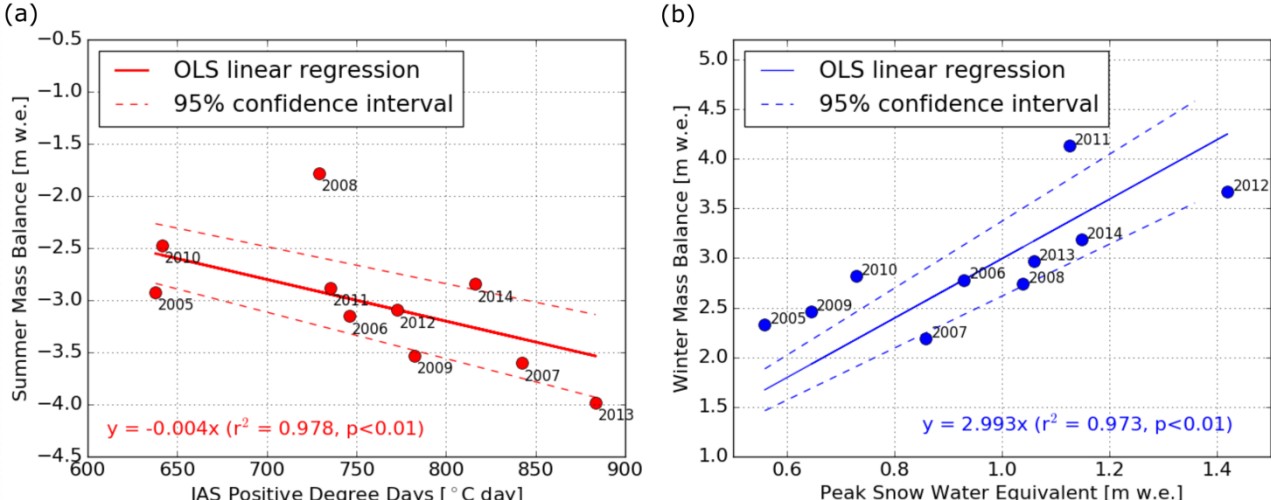

**Figure 4:** Ordinary least squares (OLS) linear regressions between climate data and glaciological mass balance for 2005-2014. (a) Summer, i.e. July, August, September (JAS), positive degree days versus summer mass balance. The linear regression is shown in red text and is plotted (solid red line). The 95% confidence interval on the regression is also shown (dotted red lines). (b) Peak snow water equivalent (SWE) versus winter mass balance. SWE data are from the Mount Allen snow course. The linear regression is shown in blue text and is plotted (sold blue line). The 95% confidence interval on the regression is also shown (dotted blue lines).

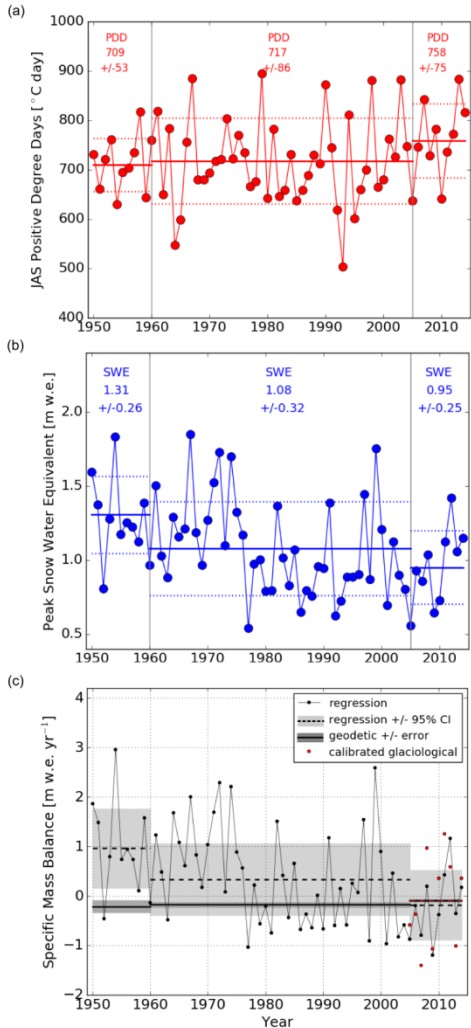

**Figure 5:** Annual climate data for 1950-2014 and comparison of regression and geodetic mass balance results. (a) Positive degree days (PDD) calculated from the Kalispell Historical Climate Network summer, i.e. July, August, September (JAS), temperature data, adjusted for elevation by the standard lapse rate. (b) Peak snow water equivalent (SWE), calculated from Mount Allen snow course data. Mean (solid line) and one standard deviation (dashed line) of PDD and SWE for each geodetic mass balance interval (1950-1960, 1960-2005, 2005-2014) are plotted and reported in text with standard deviation. (c) Regression results for annual mass balance (black dots, thin black line) are shown, as are regression results for the average mass balance from 1950-1960, 1960-2005, 2005-2014 (black dotted lines). Calibrated, annual glaciological balances are shown (red dots), as is the average of this glaciological balance from 2005-2014 (red dotted line). Errors, set by confidence intervals on the linear regression, are also shown (light gray boxes). Geodetic mass balances for 1950-1960, 1960-2005, and 2005-2014 are plotted (horizontal black lines) with errors (dark gray boxes).

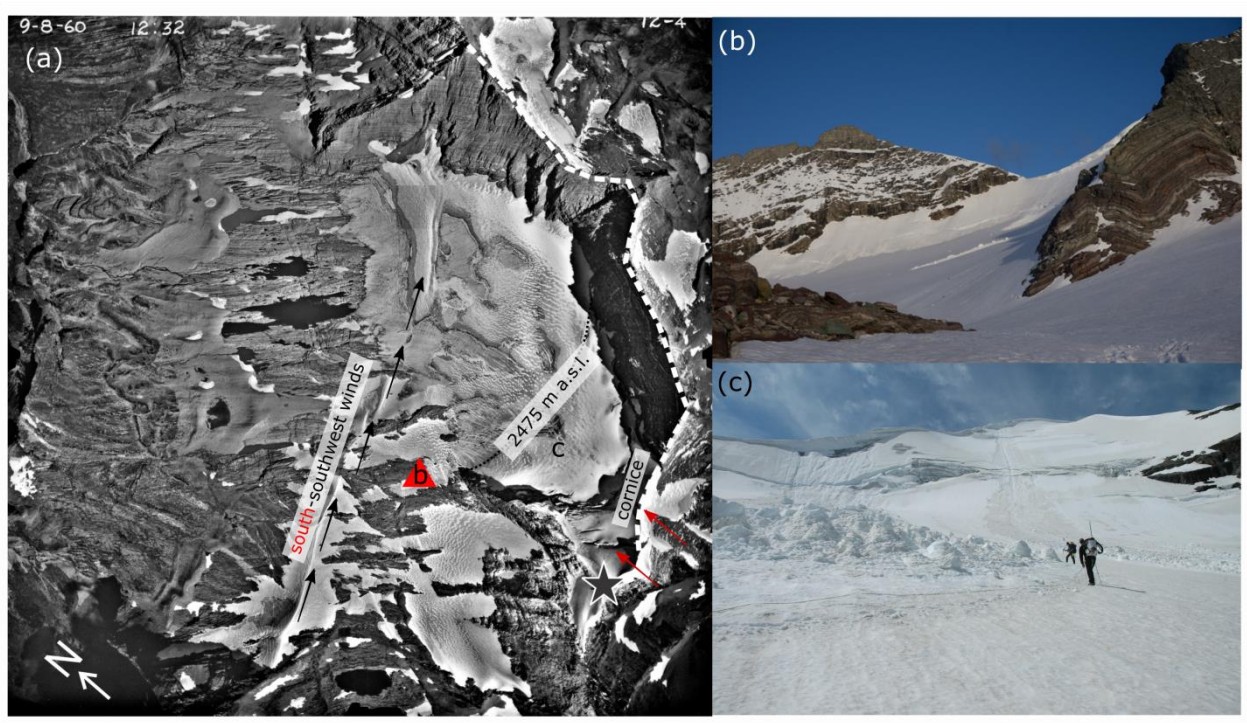

**Figure 6:** Snow avalanches and wind drifted snow at Sperry Glacier. (a) Aerial photograph of Sperry Glacier taken on 8 September 1960 shows a prominent ridge of wind drifted snow that is evidence of southwest winds in the basin (black arrows), is shown. The cornice that is evidence of south winds (red arrows) is labeled. Sperry Glacier meteorological station is shown (red triangle) and the approximate elevation (2475 m a.s.l.) above which the winter and summer mass balance gradients change is drawn. The photograph also shows the glacier lapping over the top of Gunsight Peak (black star) at the highest elevation of the ridge (dotted white line). Photograph credit: USGS Climate Change in Mountain Ecosystem photograph archives. (b) Sperry Glacier in June, 2010. Snow sluffing and loose avalanches off the headwall are depicted. Photograph was taken from the Sperry Glacier meteorological station, labelled "b" in (a). Photograph credit: Joel Harper. (c) Sperry Glacier in June, 2006. Cornice collapse (on the left) often causes a localized slab avalanche. Loose avalanche depicted on the right. Photograph credit: USGS Climate Change in Mountain Ecosystem photograph archives. Photograph was taken from the approximate location labelled "c" in (a).

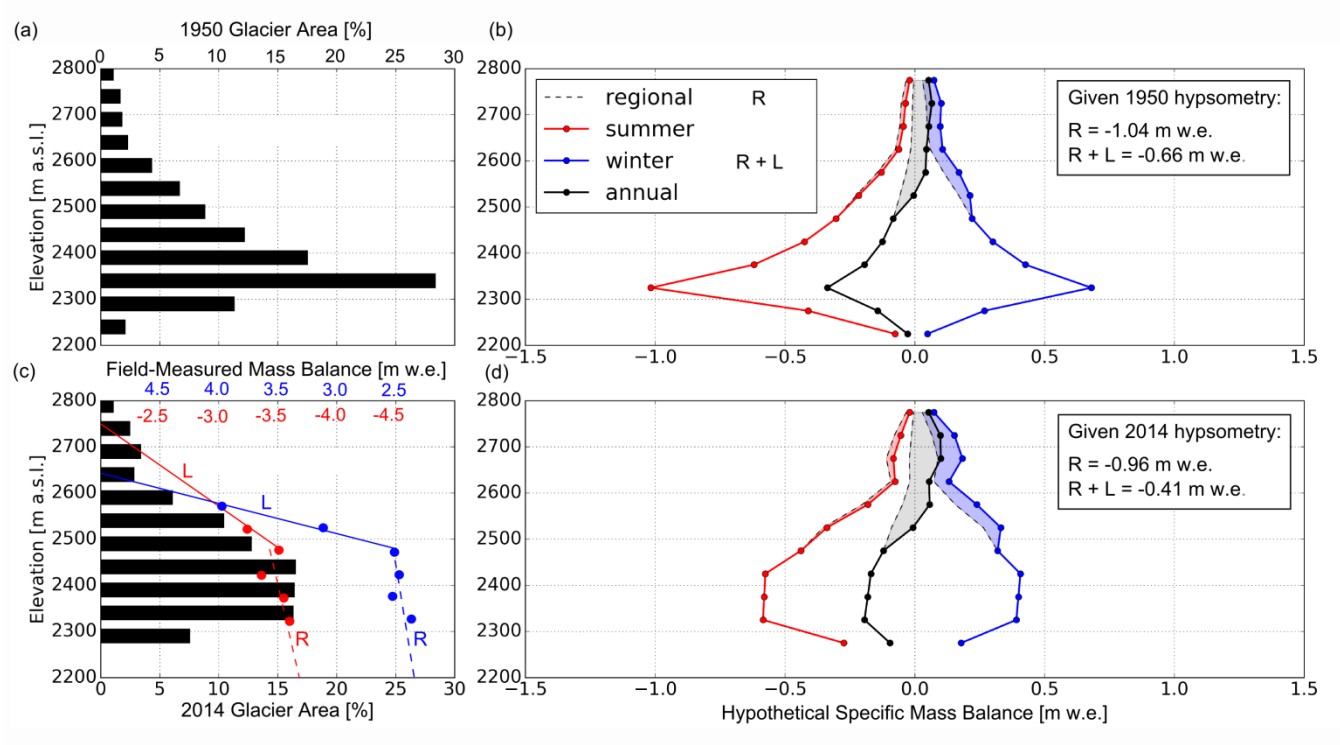

**Figure 7:** Parsing local from regional mass balance drivers. (a) Hypsometry of the 1950 glacier, presented in 50-m elevation bands (black bars). (b) Sperry Glacier hypothetical mass balance, as it varies with elevation, given 1950 glacier hypsometry and mass balance gradients. Solid lines show the summer (red), winter (blue), and annual (black) balances calculated using field-measured local (L) and regional (R) mass balance gradients shown in (c). Dotted black lines show the same but excluding L at high elevations, i.e. using R only. (c) Hypsometry of the 2014 glacier surface (black bars), and mass balance gradients measured at Sperry Glacier (red and blue dots and lines). The mean of 2005-2014 summer (red dots) and winter (blue dots) point mass balances, measured at stakes sites, are plotted against the average elevation of the 50-m elevation band wherein the stake was located. Lower elevation (< 2475 m) mass balance gradients reflect regional (R) mass balance gradients for melt (dashed red line) and snow water equivalent (SWE) (dashed blue line). Higher elevation (> 2475 m) mass balance gradients reflect local (L) mass balance gradients for melt (dotted red line) and SWE (dotted blue line). (d) Sperry Glacier hypothetical mass balance as shown in (b), but given 2014 glacier hypsometry.

**Table 1:** Area-averaged glaciological mass balance values used in this study. Winter balances are as reported by Clark et al. (2017). Summer and annual balances are the result of calibration against the 2005-2014 geodetic mass balance, accomplished following the calibration steps outlined by Zemp et al. (2013). Accumulation area ratio (AAR) as reported by Clark et al. (2017) is listed for years when the seasonal snowline was mapped.

| | Summer Balance (m w.e.) | Winter Balance (m w.e.) | Annual Balance (m w.e.) | AAR |
|---|---|---|---|---|
| 2005 | -2.92 | +2.33 | -0.59 | 34% |
| 2006 | -3.15 | +2.78 | -0.37 | - |
| 2007 | -3.60 | +2.19 | -1.41 | - |
| 2008 | -1.78 | +2.74 | +0.96 | - |
| 2009 | -3.53 | +2.46 | -1.07 | 46% |
| 2010 | -2.47 | +2.82 | +0.35 | 48% |
| 2011 | -2.88 | +4.13 | +1.25 | - |
| 2012 | -3.09 | +3.67 | +0.58 | - |
| 2013 | -3.98 | +2.97 | -1.01 | 36% |
| 2014 | -2.84 | +3.19 | +0.35 | - |

**Table 2:** Digital elevation model (DEM) specifications and glacier elevation results. Acquisition dates for the original imagery used to derive elevation data are listed.

| | DEM Source | Area (km$^2$) | Median Elevation (m) | Median Slope (°) | Median Aspect (°) |
|---|---|---|---|---|---|
| 01 Sept 1950 | USGS Topographic Map 1:6,000 | 1.30 | 2317 | 15 | 315  NNW |
| 08 Sept 1960 | USGS Topographic Map 1:6,000 | 1.23 | 2325 | 17 | 315  NNW |
| 02 Sept 2005 | National Technical Means Imagery | 0.86 | 2425 | 22 | 326 NW |
| 07 Sept 2014 | National Technical Means Imagery | 0.80 | 2444 | 24 | 326 NW |

**Table 3:** Geodetic mass balance results. Net changes in volume (ΔV) and mass (ΔM) on Sperry Glacier from 1950-1960, 1960-2005, and 2005-2014 are listed. Uncertainties due to elevation error (EΔh) and density (Eρ) are listed, as are total errors ($E_t$). Geodetic mass balances (dH $dt^{-1}$) are listed with uncertainty defined by total error, expressed as a rate.

| | Net Volume Change | Net Mass Change | Elevation Error | Density Error | Total Error | Geodetic Mass Balance |
|---|---|---|---|---|---|---|
| | ΔV | ΔM | EΔh | Eρ | $E_t$ | dH $dt^{-1}$ |
| | ($m^3$ x $10^6$) | (kg x $10^9$) | (m w.e.) | (m w.e.) | (m w.e.) | (m w.e. $yr^{-1}$) |
| 1950-1960 | -3.33 | -2.83 | 1.20 | 0.18 | 1.22 | -0.22 $\pm$ 0.12 |
| 1960-2005 | -11.3 | -9.68 | 2.16 | 0.68 | 2.27 | -0.18 $\pm$ 0.05 |
| 2005-2014 | -0.90 | -0.76 | 0.26 | 0.06 | 0.27 | -0.10 $\pm$ 0.03 |

**Supplement contents:**

- Details on the glaciological mass balance calibration

- Figure S1

- Figure S2

- Figure S3

- Table S1

- Table S2

- Table S3

**Glaciological mass balance calibration**

We defined the annual glaciological balance as the sum ($B_{a.sum}$) of the site index summer ($B_s$) and winter ($B_w$) balances reported by Clark et al. (2017). We calibrated annual and summer balances using the following steps, defined in the calibration section of the Zemp et al. (2013) reanalysis procedure.

5    First, we calculated the centred glaciological balance ($\beta_t$), which is the deviation from the mean glaciological balance ($\bar{B}_{a.sum}$) for 2005-2014. This term captured interannual variability documented by glaciological data:

$$\beta_t = B_{a.sum} - \bar{B}_{a.sum} \tag{S1}$$

Next, we used this centered glaciological balance and the geodetic mass balance ($\bar{B}_{a.geodetic}$) to calculate the calibrated annual balance ($B_{a.cal}$):

10    $$B_{a.cal} = \beta_t + \bar{B}_{a.geodetic} \tag{S2}$$

Finally, we calculated the calibrated summer balance:

$$B_{s.cal} = B_{a.cal} - B_w \tag{S3}$$

These calibrated summer and annual balances are reported in Table 1.

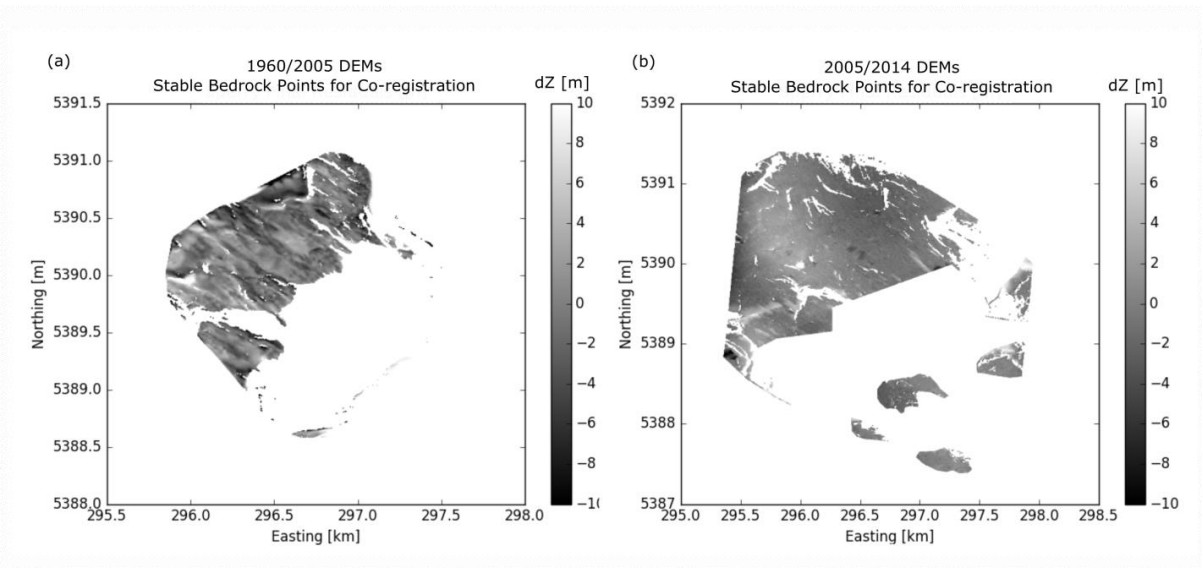

**Figure S1:** Elevation differences over the stable bedrock used for DEM co-registration are shown on a forced -10 m to 10 m scale. The full suite of elevation differences over stable bedrock terrain is shown in Figure S2.

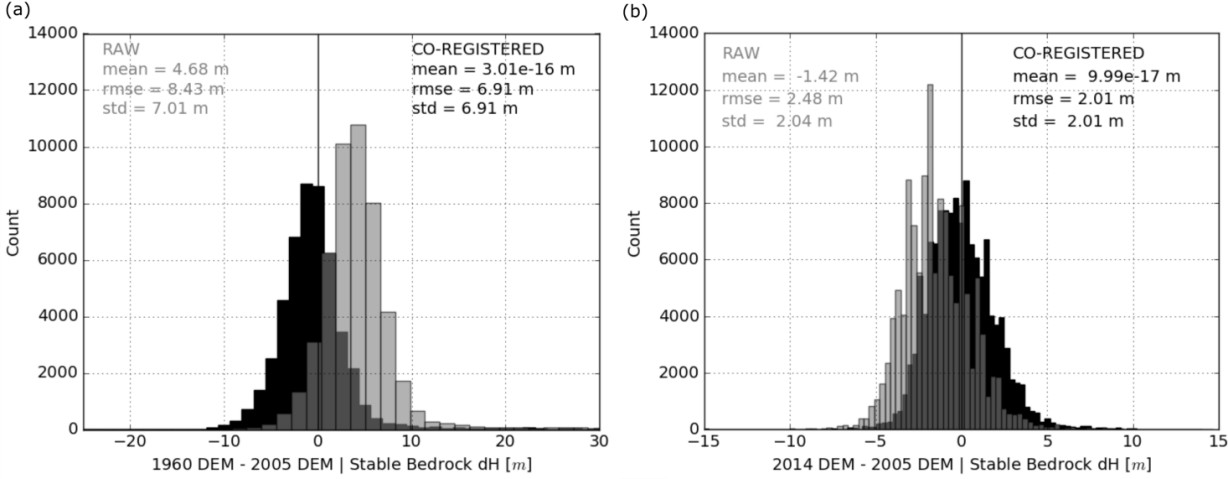

5 **Figure S2:** Elevation differences over stable bedrock terrain before (gray bars) and after (black bars) co-registration of the (a) 1960/2005 and (b) 2005/2014 DEMs. Plots show the magnitude (x-axis) and pixel count (y-axis) of elevation differences over stable bedrock terrain that is low-sloping ($< 30°$), and free of vegetation, snow, and ice. Text on the plots reports mean, root mean square error (rmse), and standard deviation (std) of these elevation differences for the raw (gray text) and co-registered (black text) results. The number of pixels used for co-registration for 1960/2005 (n = 23,951) was fewer than for
10 2005/2014 (n = 127,440) due to the 2005 and 2014 DEMs covering more bedrock terrain. Dark gray bars are an artefact of light gray bar transparency.

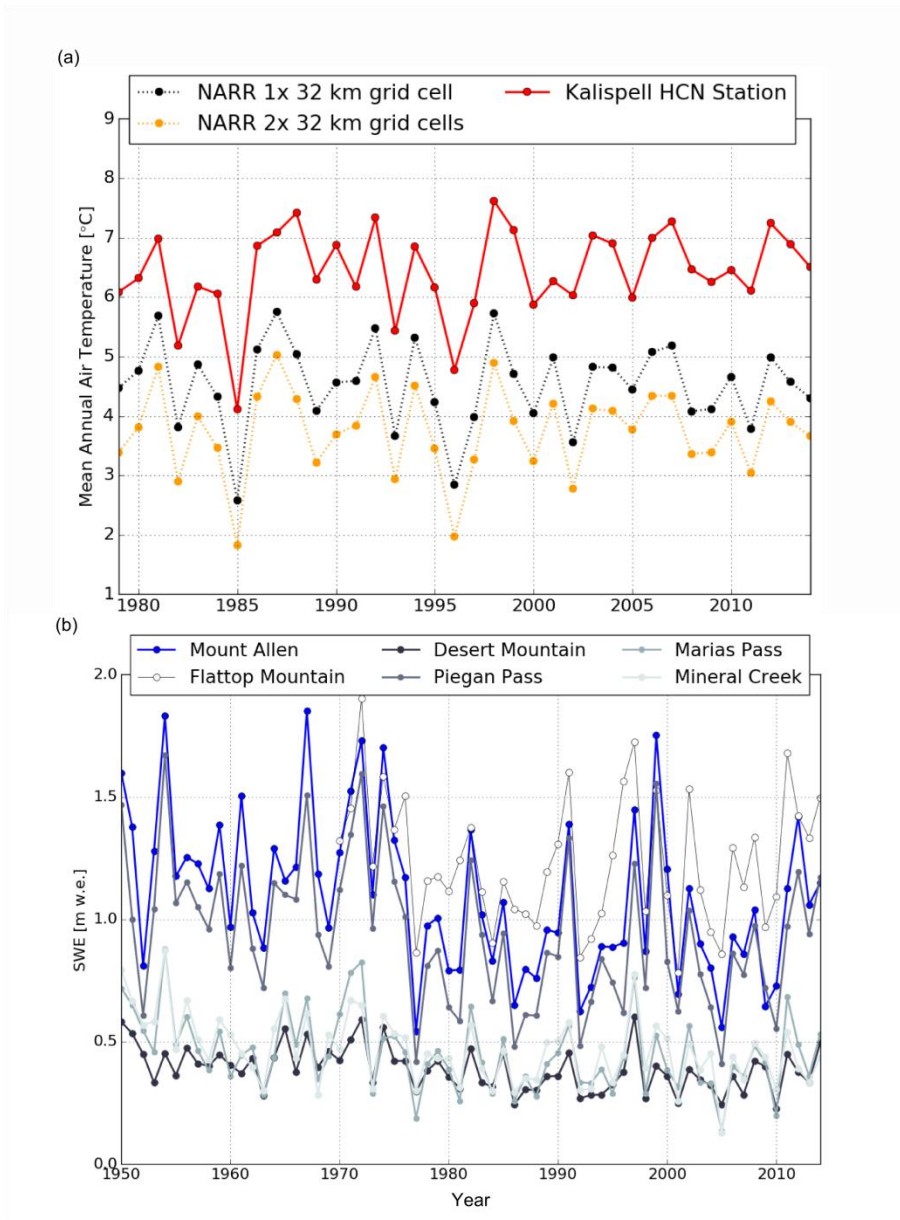

**Figure S3:** Auxiliary meteorological and snow data. (a) Mean annual air temperatures measured (solid lines) at the Kalispell Historical Climate Network (HCN) station (red) compared with mean annual air temperatures according to North American Regional Reanalysis output (dotted lines) for a single 32-km grid cell containing the Kalispell station (black), and two 32-km grid cells containing the Kalispell station and Sperry Glacier (yellow). (b) Annual peak SWE data for six snow stations in the study area. Station details listed in Table S3.

**Table S1:** Digital elevation model (DEM) results for the truncated glacier. Differences from the full glacier results (see Table 2), where 2014 elevations were used to fill in the missing upper section, are reported in italics.

|  | Area<br>($m^2$ x $10^6$) | Median Elevation<br>(m a.s.l.) |
|---|---|---|
| 1950 DEM | 1.16 | 2305 |
|  | *-0.14* | *-12* |
|  | *-11%* | *-0.5%* |
| 1960 DEM | 1.10 | 2310 |
|  | *-0.13* | *-15* |
|  | *-12%* | *-0.7%* |

**Table S2:** Geodetic mass balance results for the truncated glacier. The net change in volume ($\Delta V$) and mass ($\Delta M$) are listed, as are mass change rates ($dH\ dt^{-1}$). Differences from full glacier results, where 2005-2014 values were used to fill in the missing upper section, are reported in italics.

| | Net Volume Change | Net Mass Change | Mass Change Rate |
|---|---|---|---|
| | $\Delta V$ | $\Delta M$ | $dH\ dt^{-1}$ |
| | $(m^3 \times 10^6)$ | $(kg \times 10^9)$ | $(m\ w.e.\ yr^{-1})$ |
| 1950-1960 | -3.54 | -3.00 | -0.26 |
| | *-0.21* | *-0.17* | *-0.04* |
| | *-6%* | *-6%* | *-18%* |
| 1960-2005 | -12.3 | -10.5 | -0.21 |
| | *-1.00* | *-0.82* | *-0.03* |
| | *-9%* | *-9%* | *-17%* |

**Table S3:** Snow data that were assessed, but not used in the linear regression. The type of snow measurement site and correlation coefficients ($r^2$) between glaciological winter mass balance and peak SWE measured at the snow measurement site for 2005-2014 are listed. Winter proportionality factors ($m_w$), calculated using 2005-2014 glaciological winter balance data; map distance from Sperry Glacier, snow measurement site elevation, and start of the historical record of the snow data (First Year) are also listed. The dash (-) indicates not measured.

| Site | Measurement | $r^2$ | $m_w$ | Distance | Elevation | First Year |
|---|---|---|---|---|---|---|
|  |  |  | (unitless) | (km) | (m) |  |
| Sperry Glacier | - | 1 | 1 | 0 | 2450 | - |
| Flattop Mountain | SNOTEL | 0.990 | 2.32 | ~15 | 1920 | 1970 |
| Mount Allen | Snow Course | 0.973 | 2.99 | ~14 | 1737 | 1922 |
| Desert Mountain | Snow Course | 0.960 | 7.93 | ~27 | 1707 | 1937 |
| Piegan Pass | Snow Course | 0.959 | 3.28 | ~14 | 1676 | 1922 |
| Marias Pass | Snow Course | 0.939 | 6.68 | ~46 | 1600 | 1934 |
| Mineral Creek | Snow Course | 0.945 | 7.24 | ~13 | 1219 | 1939 |