# Peer review of "Local topography increasingly influences the mass balance of a retreating cirque glacier"

_The Cryosphere, 2018_

## Referee Comment (RC1) · Anonymous Referee #1 · 9 Apr 2018

Review of Florentine et al. - Local topography increasingly influences the mass balance of a retreating cirque glacier (09/04/2018)

Overview ————- The authors present an analysis of the mass balance of a small mountain glacier in North America with respect to regional climatic trends and the influences of local topography. Using a combination of geodetic and glaciological approaches, the authors observe a small reduction of average mass loss in the glacier between three periods of time, namely 1950-1960, 1960-2005 and 2005-2014, despite being strongly distinct to regional mass balance estimates, based upon positive degree days and winter precipitation records since 1950. The paper is concise and very well

written with a good amount of thought given to potential sources of error and uncertainty. The topic is relevant to the cryospheric community given the decreasing size of many mountain glaciers and the increased role of local factors on the glacier-wide mass balance. I believe the manuscript to be of sufficient quality and relevance for publication in The Cryosphere and think that the work should be published.

Nevertheless, I believe the authors need to address more the role of the local affects in promoting the slow down of mass loss for a glacier such as Sperry Glacier. The manuscript does well to highlight the assumptions of regional climate and the stark contrasts with the local mass balance, but needs to further highlight the role of local events and their spatial domain, so that such processes may be more readily compared with the observations of mass balance across the glacier. The authors present a nice case study of the glacier with a likely increase in the contribution from local factors, though I believe the information on local factors needs to be quantified more and presented more clearly for the reader on maps that can be interpreted. Various figures do not contribute enough the arguments of the manuscript and thus limits the conclusions that are drawn to some extent. The results presented on regional mass balance estimates vs. the geodetic changes imply that, although local topography/meteorological conditions is likely having a greater influence on mass balance, Sperry Glacier has, since 1950 at least, been decoupled from the regional climate.

One of the largest drawbacks of the paper is the lack of introduction and, importantly, discussion of the literature to contextualise the findings in the Glacier National Park. For a revision of this manuscript, I believe more needs to be added to compare Sperry Glacier with regional vs. local influences as published elsewhere in the world. I provide some suggestions of relevant research that could be included here and specific questions and comments that should be addressed before publication in the journal. I consider, on the whole, a minor revision, though would like to see some larger changes to the introduction, figures and discussion of the local effects on the glacier.

Abstract ——— P1.L17 ". . .closely predicts the geodetically measured mass loss from

2005-2014." Please quantify 'close'.

P1.L18-20 Overestimates 1950-60 Mass balance – does this not imply that recent mass balance (2005-2014) can be better explained by regional climate? See later comment regarding interpretation of the results with respect to local vs. regional influences.

Intro —— P1.L23 Yes, radiation inputs are dominant in summer for most glaciers, but I think it would be appropriate to rephrase this as radiation and air temperature driven (this can often be the case for coastal environments/ maritime glaciers where longwave and turbulent fluxes can sometimes dominate) – Also because you relate mass balance to PDDs in this paper (though the PDDs and shortwave radiation are, of course, well related).

P1.L27 What about Carturan et al. (2012) and the case of Italy's lowest elevation glacier, Montasio Occidentale? I think this should be included in your introduction. Also, what about the variability in meteorological conditions and energy balance processes for a small glacier such as shown by Hannah et al. (2000)? Some mention to the changing role of glacier energy balance would be good.

P2.L2-3 Hoffman 2007 found no trend with winter precipitation in the 20th Century. I think you should add more details about the glacier sizes during that study. Was there a decline in winter precipitation for their study? How else can you state the role of local processes in your following sentence? In your manuscript, you are talking about increased sensitivity to regional climate for your earlier periods of observation (during 20th Century). How will Hoffman's findings compare to your results? I think this is something to be discussed later in your work.

P2.L8-10 I think a reference here would be suitable

P2.L10-14 Again some more reflection on Hoffman's findings would be good for this introduction. Is Andrews Glacier facing south? Are the radiation loads or some other

factor sufficient to explain why regional climate still outweighs the local influences? In addition, what about the controls of the atmospheric boundary layer adjustment for the retreat of a mountain glacier? For example, one could expect a diminishing katabatic boundary layer for fragmenting glaciers (see Carturan et al. 2015 - TC) and an increase in the sensitivity of the near-surface air to air temperature fluctuations outside the boundary layer (see Greuell and Böhm, 1998 and Shaw et al., 2017 – JoG).

Study site ——— P2.L24 Please give coordinates centred on the glacier

P3.L3 Historical photographs from which year?

Methods ——— P3.L27 Can you provide more information about the area size and locations on 'stable' ground that was used for the co-registration. Was there presence of landslips/rockfalls that may account for some of the large vertical differences? Consider including a figure to the supplementary information.

P4.L3 What aerial imagery?

P4.L11 Can you elaborate here why you decided to apply the Huss Approximation for glacier ice density and not use the values provide from the glacier itself in the case of Clark et al., 2017 with an error range?

P4.L25 How does the quantity of stable bedrock, to provide more certainty in elevation differences, vary with elevation in your study site?

P5.L23 Can you also give a bit more detail about how this information was filled? You mean to say that (as with lines 26-27) that you used the 2014 generated DEM elevations for these areas that were missing? You state that the rate of mass change (1950-2014) is the same... but you mean to say there is no mass change, as you use the same DEM values? Or there is no DEM information for these years, but just a rate of change for 2005-2014 applied? Perhaps I am missing something clear with this paragraph, though perhaps you can state it more directly? Also, including the elevations that were missing from the 1950-60 photographs and a map of the missing areas with the supplementary information would be valuable for interpretation of your results, despite that you state the differences are small and within the error margins.

P6.L10 Please provide further details on the calibration of the biases in raw mass balance data on Sperry Glacier here. Please plot the mass balance measurements (I assume stake data?) in the map of Figure 1b.

P6.L15-16 A very minor point, but can SWE be considered a meteorological variable ('meteorological data')?

P6.L20-22 I do not argue with the use of Kalispell to represent PDDs for this analysis, though are there no other meteorological information within the greater area that may aid construction of ensemble PDDs for 'regional' climate? Is there any indication of local processes at Kalispell that may obscure the relationships you derive? Furthermore, are there any off-glacier meteorological observations in the basin of the study site that give indication to the representation of PDDs derived from Kalispell?

P6.L23 Partly related to the previous point, how did you 'vet' the Kalispell data record using the Sperry Glacier one? The application of an above-ice AWS would reveal not only a reduced average temperature compared to that outside the thermal influence of the glacier, but also a 'dampened' diurnal cycle (again see Carturan et al., 2015- TC).

P6.L26 Did you utilise only JJA to correct the temperature using the vertical temperature lapse rate? See above point. Where was the Sperry AWS? Plot also in Figure 1b.

P7.L11 State which observations specifically were used for this analysis.

P8.L3 Is there evidence from the glaciological observations on Sperry Glacier to suggest that the melt season does not (in a 'typical year') begin earlier than July?

P8.L10-12 I think the information on your assessment of the local processes (i.e. avalanching and snow drifts) are lacking in this section. Again, information about where

observations from Clark et al. (2017) are in relation to the qualitative information used would be highly informative for the reader to comprehend the arguments about the increased role of local conditions/processes in governing the mass balance of Sperry Glacier.

Results ——— P9.L2-3 Is there any information on the development of debris-covered ice at the terminus of the glacier, which may enhance or diminish local melt rates?

P9.L26-27 Perhaps I miss something here, but 1950-60 and 2005-14 don't seem all that comparable in Table 3. Admittedly there are not huge differences... but still statistically significantly different, no?

P10.L4-7 This is a nice evaluation of potential energy losses. Could you provide the values in Wm-2 as would be more typically reported for studies for glacier energy balance? Is the 0.36 m w.e. deficit for the whole glacier over the whole period?

P10.L10-13 Can you indicate what evidence you have for wind effects on glacier mass balance? Evidence from Figure 8 show a potentially important source of local mass input, though your reported results here don't strongly support the information you have already presented. Figure 8 does very little alone to bring together your ideas as it does not contain information relating to the elevation ranges of the wind drift snow and thus, the values of wind scour/accumulation effects in the following sentence suggest large spatial variations in mass balance, but do little more. Perhaps you could provide data on typical wind direction for the basin, based upon the Sperry Glacier AWS records. Do these tie in with more recent wind deposits apart from what was seen in the historic images? Is there a pattern here that might explain the mass balance trends which don't conform to a regional mass balance assumption? I think compiling a more useful figure with combinations from Figure 7 and 8 with a digitised map could be appropriate. Furthermore, could you provide information on the exposure of the Sperry Glacier by topographic information from your DEM? For example, the 'Sx' parameter following Winstral et al. (2002) could be appropriate for looking at the potential effects of wind

on the initial deposition of snow in the winter mass balance and the re-distribution in summer. Is there anything to suggest these effects have changed through the decades, or just the reduction of glacier ice at lower elevations slowing the total glacier mass loss? A full-scale analysis would be beyond the scope of the paper, though I believe more is required here to argue to the case of local vs regional effects.

P10.L17-18 Again, this gradient really argues the case for avalanching of material, though information about the location of the observations would be both useful and interesting.

Discussion ————— P10.L30 I think it could be argued, particularly based upon your results of Figure 6, that even since 1950, the small size and topographical characteristics of Sperry Glacier have limited its mass loss. A 'small glacier' is relative to whom you ask, though in my experience, 3.24km2 (the size of the glacier in the historical images) is small. The key aspect of this paper suggests that further retreat has increased the role of local topography. The evidence from geodetic change, which has been well assessed with regards to errors, would suggest that, indeed, the glacier mass loss has slowed despite conditions favouring its demise. Nevertheless, a point to make here is that, compared to what the regional trends in PDDs and winter accumulation would suggest, the glacier-wide mass balance of Sperry Glacier has since 1950 (and likely further back in time) been somewhat decoupled from what the regional climate would prescribe. One may have to assess a much more historic form of the glacier, with much greater size, to identify a stronger relationship to regional climatic trends.

P11.L8 Why is there a sudden steepening of the summer mass balance for the top of the glacier? Is there not just a lower mass balance record between the ∼2420 and 2550 values in Figure 9? What can explain this?

P11.L10-17 I think I miss something with regards to the representation of information in this figure (10). The regional 'lapse rates' (a term I would consider changing to mass balance gradient) is that derived from Figure 5, I presume? However, for which periods

of time are these mass balances shown, the complete period? Maybe the reflection of mass balance relating to glacier hypsometry needs some further clarification here.

P11.L19 79% of what, exactly? Is this evidence shown in Figure 10? To me, these contributions of summer and winter do not show such a strong difference as 79:21.. Again, please provide greater clarification to the reader.

P11.L26 The findings of Mattias Huss are highly relevant in this discussion, though I would like to see more of this discussion section related to the information which is in (and should be added to) the introduction. See comments on introduction section for suggestions on linking your discussion more with past work on small mountain glaciers.

Figures ——— Generally, the figures are well presented and clear. I have a few suggestions about combining figures and using others to a greater effect:

Figure 1b should display the locations of the mass balance observations for reference of the reader.

Figure 2 could be combined with Figure 1 and set as a larger typeset figure for the article. The map information in both is relevant to the location of things in your study basin which can be referred to many times to aid your conclusions.

Figure 3: Can you change the elevation interval averages and error bars (light green) to a different colour? Perhaps red?

Figure 4: I think there is perhaps a bit too much information contained within the caption which is sufficiently explained in the text.

I think it could be useful to combine Figures 5 and 6 and stack them vertically as they show similar information and time-series trends. The results of Figure 6 are clearly very strongly influenced by the winter precipitation in Figure 5 and combining these figures would aid interpretability.

As mentioned in the review, Figure 7 and Figure 8 provide some potentially useful information, though it is not used to great effect. Some information about the locations, elevations and extent of these observations (where possible) would be useful in comparison to the digitised glacier map of the same (or approx.) time period. This should be leveraged to explain some of the mass balance behaviour of Sperry Glacier in your study. As suggested previously, some information about the consistency of wind drifts and/or wind scouring (and a better map of the location(s) could be useful). For example, how could we then treat these effects in future modelling efforts to better represent the future of small glaciers? Should we/could we at all?

Figure 9: I think this figure is very important to the suggestion of localised processes, particularly during winter. However, it is not clear to me, the derivation of regional and local 'lapse rates' here. Again, I would suggest an alternative term for this too, perhaps mass balance gradient.

Figure 10: Again, I think this a good and valuable figure to the paper. I'm not convinced that the text explains the information of the figure fully, or perhaps I miss something. Please try and make this clearer to the reader. For example, which are the gradients (here referred to as gradients) which are mapped onto the glacier hypsometry in these two years?

Supplementary information ——————————————- I'm not completely sure what Figure S2 is showing to support your work here and appears to only show a corrected result despite being intended to show a pre- and post-correction.

---

## Referee Comment (RC2) · Anonymous Referee #2 · 11 Apr 2018

**Review** to Florentine et al. (2018): *Local topography increasingly influences the mass balance of a retreating cirque glacier*, submitted to The Cryosphere.

**General**

The authors present a study of glaciological and geodetic mass balance estimates from Sperry Glacier, a small cirque glacier in Glacier National Park, Montana, USA. Modelling of past surface mass balances applying a statistical model, which was calibrated with recent mass balance observations, yields a bias between the modelled surface mass balance and observed geodetic mass balances. The authors interpret this bias as an increased control of local climatic mass balance drivers and a decreased regional climatic influence.

The study is a decent example of the climate proxy potential of small glaciers and is therefore a valuable contribution to the journal. However, the key findings do not completely exclude alternative interpretations and thus, I'd like to address three concerns, which may sum up to a major revision.

1. Quality of SWE data (section 3.5.1): Mount Allen SWE measurements serve as input to the regression model. Hence the data quality is decisive for interpreting model results. From visual analysis Figure 5b suggests a step change of peak SWE on Mount Allen in the mid 1970s. Does this change also appear in the other SWE observations listed in Table S3 or is it a local effect or an inhomogeneity of the data series?
2. Kalispell air temperature data: Assuming one climate station in ~50 km distance to the glacier representing the regional climate needs more justification. How do the summer temperatures compare to e.g. the NCEP North American Regional Reanalysis (to the closest grid point to Sperry Glacier or to Kalispell climate station)?
   Why is the ablation season confined to the summer months JAS? Degree-day approaches are based on the correlation of ablation to positive air temperature sums (parametrizing the available energy for melt) over the whole ablation season or even the whole year (e.g. Hock, 2003). How would the results of the mass balance regression change using air temperatures for the whole year or at least for the months MJJAS instead of JAS?
3. Discussion of the mass balance regression and the interpretation of the increase of local topography to the mass balance: The regression model is based on two proportionality factors $(m_s, m_w)$. The discussion implicitly presumes both factors constant over time, but this needs to be addressed more comprehensively.
   - Winter proportionality factors might be variable (Galos et al., 2017; Huss et al., 2008), causing random errors, but systematic errors due to changes in large and meso scale atmospheric flow patterns (Huss et al., 2010) will alter the mass balance regression and weaken the argument of the increasing influence local topography to the mass balance. Can the authors exclude systematic changes of $m_w$?
   - Degree-day factors (i.e. the summer proportionality factor) are not constant over time, if the glacier surface area is largely changing. The major reason for this systematic change of the degree-day factor is the albedo feedback (e.g. Naegeli and Huss, 2017). An approximation to this feedback is the change of accumulation area ratio (AAR). Results of the geodetic survey show a thickening of the accumulation area and a concurrent glacier retreat, which means that the accumulation area remained rather constant, while the glacier lost wide parts of its ablation area. Hence, in relation to the total glacier area the AAR increased, resulting in higher mean albedo of the glacier and thus a lower degree-day

factor. This effect would indeed strengthen the finding of the increasing influence local topography to the mass balance. (Interestingly, in all studies I'm aware of, the albedo feedback increases melt because the glaciers generally lose their accumulation areas due to rising equilibrium line altitudes.)

**Introduction**

This chapter must more elaborate on the peculiarities and the definition of a small glacier, which in the manuscript seems to be synonymous to a cirque glacier and a very small glacier. Whatever classification is used, the important message in this study is that the local topography has a high influence on the accumulation regime of the Sperry Glacier. In the last paragraph the authors describe briefly the areal change of Sperry Glacier and formulate their research question. At this point I suggest introducing the term accumulation area ratio (AAR) (Cogley et al., 2011) as it (i) presumably describes that the glacier lost its ablation area while the accumulation area almost remained constant and (ii) this fact is crucial to interpret the findings later in the manuscript.

**Methods**

I suggest adding a paragraph that the two mass balance methods used (geodetic and glaciological) consider different processes of mass change (e.g. Klug et al., 2018; Zemp et al., 2013). In this study the differences will presumably be smaller than the given errors. The exclusion of methodological differences will support the discussion of the regression model later in the manuscript.

In Eq. 1 the authors derive the geodetic mass balance based on the initial glacier area. By convention (Cogley et al., 2011), the mean area between initial and final state is used (Andreassen et al., 2016; Klug et al., 2018; Lang and Patzelt, 1971; Zemp et al., 2013 etc.), thus a recalculation of the geodetic mass balance is required.

**Results**

Section 4.3 supports the albedo feedback mentioned above.

**Figures & Tables**

Figure 1a: Explain HCN.

Figure 1b: Give exact date of the aerial image. Add the location of the mass balance stakes and the meteorological station.

Figure 4b: Indicate source of the peak SWE data.

Figure 6: Add the glaciological mass balance values.

Figure 10: Add labels a-d.

Figure S2: What is the added value of this Figure? The grey line is hardly visible.

Figures 5, 6, S2: Figures do not depict continuous data as suggested by the x-axis. Use a bar chart instead of a line graph.

Table 1: Add columns of ELA and AAR.

Table 2 and S2: As glacier hypsometry is generally not normally distributed the median elevation is preferable to the mean elevation.

Table S3: Explain $m_w$ in the table capture.

**Specific comments**

P 1, L 15 and elsewhere in the manuscript: Are negative mass loss rates a mass gain? This is pedantic but I suggest using neutral formulations that do not compete with the sign of the corresponding value.
Glaciologists traditionally use m w.e. as mass balance unit. Let's convert to kg/m² (≡ mm w.e.), because this is the SI unit and outside the glaciological community nobody understands w.e.

P 2, L 20: relative to what?

P 6, L 26: The air temperature lapse rate must be a negative value.

P 6, L 27: Rephrase the sentence. 7 years ago is not recently.

P 7, L 1: Was the glacier surface on average 35 m lower in elevation or just the terminus?

P 7, L 4: Change the units to °C and maybe rephrase the last part to "..., and cumulatively -24°C and -14°C changes to PDD."

P 9, L 5+6: Replace million and billion by $10^6$ and $10^9$, respectively.

P 9, L 9: Link to Table 3?

P 9, L 28: Rephrase the sentence beginning with "Enough years are positive…"

P 10, L 4: Explain how you derived this number.

P 11, L 1: Explain the meaning of uniform mass balance gradients. Gradients in the ablation area are usually different from those in the accumulation area, mainly because of the higher albedo in the latter (Kaser et al., 1996; Kuhn et al., 1999).

**References**

Andreassen, L. M., Elvehøy, H., Kjøllmoen, B. and Engeset, R. V.: Reanalysis of long-term series of glaciological and geodetic mass balance for 10 Norwegian glaciers, Cryosphere, 10(2), 535–552, doi:10.5194/tc-10-535-2016, 2016.

Cogley, J. G., Hock, R., Rasmussen, L. A., Arendt, A. A., Bauder, A., Braithwaite, R. J., Jansson, P., Kaser, G., Möller, M., Nicholson, L. and Zemp, M.: Glossary of Mass Balance and Related Terms, UNESCO-IHP, Paris., 2011.

Galos, S. P., Klug, C., Maussion, F., Covi, F., Nicholson, L., Rieg, L., Gurgiser, W., Mölg, T. and Kaser, G.: Reanalysis of a 10-year record (2004–2013) of seasonal mass balances at Langenferner/Vedretta Lunga, Ortler Alps, Italy, The Cryosphere, 11(3), 1417–1439, doi:10.5194/tc-11-1417-2017, 2017.

Hock, R.: Temperature index melt modelling in mountain areas, Journal of Hydrology, 282(1–4), 104–115, doi:10.1016/S0022-1694(03)00257-9, 2003.

Huss, M., Bauder, A., Funk, M. and Hock, R.: Determination of the seasonal mass balance of four

Alpine glaciers since 1865, Journal of Geophysical Research, 113, F01015, doi:10.1029/2007JF000803, 2008.

Huss, M., Hock, R., Bauder, A. and Funk, M.: 100-year mass changes in the Swiss Alps linked to the Atlantic Multidecadal Oscillation, Geophysical Research Letters, 37(10), 1–5, doi:10.1029/2010GL042616, 2010.

Kaser, G., Hastenrath, S. and Ames, A.: Mass balance profiles on tropical Glaciers, Zeitschrift für Gletscherkunde und Glazialgeologie, 32(2), 75–81, 1996.

Klug, C., Bollmann, E., Galos, S. P., Nicholson, L., Prinz, R., Rieg, L., Sailer, R., Stötter, J. and Kaser, G.: Geodetic reanalysis of annual glaciological mass balances (2001-2011) of Hintereisferner, Austria, The Cryosphere, 12, 833–849, doi:10.5194/tc-12-833-2018, 2018.

Kuhn, M., Dreiseitl, E., Hofinger, S., Markl, G., Span, N. and Kaser, G.: Measurements and Models of the Mass Balance of Hintereisferner, Geografiska Annaler, Series A: Physical Geography, 81(4), 659–670, doi:10.1111/1468-0459.00094, 1999.

Lang, H. and Patzelt, G.: Die Volumenänderung des Hintereisferners (Ötztaler Alpen) im Vergleich zur Massenänderung im Zeitraum 1953 - 1964, Zeitschrift für Gletscherkunde und Glazialgeologie, 7(1–2), 39–55, 1971.

Naegeli, K. and Huss, M.: Sensitivity of mountain glacier mass balance to changes in bare-ice albedo, Annals of Glaciology, 58, 119–129, doi:10.1017/aog.2017.25, 2017.

Zemp, M., Thibert, E., Huss, M., Stumm, D., Rolstad Denby, C., Nuth, C., Nussbaumer, S. U., Moholdt, G., Mercer, A., Mayer, C., Joerg, P. C., Jansson, P., Hynek, B., Fischer, A., Escher-Vetter, H., Elvehøy, H. and Andreassen, L. M.: Reanalysing glacier mass balance measurement series, The Cryosphere, 7(4), 1227–1245, doi:10.5194/tc-7-1227-2013, 2013.

---

## Author Comment (AC1) · 18 May 2018

**Review** to Florentine et al. (2018): *Local topography increasingly influences the mass balance of a retreating cirque glacier*, submitted to The Cryosphere.

Author responses in blue.

**General**

The authors present a study of glaciological and geodetic mass balance estimates from Sperry Glacier, a small cirque glacier in Glacier National Park, Montana, USA. Modelling of past surface mass balances applying a statistical model, which was calibrated with recent mass balance observations, yields a bias between the modelled surface mass balance and observed geodetic mass balances. The authors interpret this bias as an increased control of local climatic mass balance drivers and a decreased regional climatic influence.

The study is a decent example of the climate proxy potential of small glaciers and is therefore a valuable contribution to the journal. However, the key findings do not completely exclude alternative interpretations and thus, I'd like to address three concerns, which may sum up to a major revision.

We appreciate your resolute comments and have revised the manuscript accordingly.

1. Quality of SWE data (section 3.5.1): Mount Allen SWE measurements serve as input to the regression model. Hence the data quality is decisive for interpreting model results. From visual analysis Figure 5b suggests a step change of peak SWE on Mount Allen in the mid 1970s. Does this change also appear in the other SWE observations listed in Table S3 or is it a local effect or an inhomogeneity of the data series?

The mid 1970s step change appears in other SWE observations. We have defended the quality of SWE data by adding text, a supplementary figure, and references that speak to the regional consistency of this step change and its likely cause:

(now P10.L18) "The step change in peak SWE during the mid-1970s is consistent with other regional SWE records (Fig. S3b, Supplement), and has been interpreted as a result of a modal change in the Pacific decadal oscillation, a pattern of ocean climate variability which is closely tied to peak SWE in this region (McCabe and Dettinger, 2002; Pederson et al., 2011; Selkowitz et al., 2002)."

2. Kalispell air temperature data: Assuming one climate station in ~50 km distance to the glacier representing the regional climate needs more justification. How do the summer temperatures compare to e.g. the NCEP North American Regional Reanalysis (to the closest grid point to Sperry Glacier or to Kalispell climate station)?

To justify our assumption, we added analysis of North American Regional Reanalysis output:

(now P6.L30): "Temperatures measured at Kalispell are highly representative of regional climate as reflected by gridded values from North American Regional Reanalysis output (Supplement, Fig. S3a)."

Why is the ablation season confined to the summer months JAS? Degree-day approaches are based on the correlation of ablation to positive air temperature sums (parametrizing the available energy for melt) over the whole ablation season or even the whole year (e.g. Hock, 2003). How would the results of the mass balance regression change using air temperatures for the whole year or at least for the months MJJAS instead of JAS?

We added text to explain our confinement of the melt season to JAS, and describe the effect of instead considering MJJAS:

(now P7.L8) "We follow the convention defined by previous GNP melt modelling (Clark et al., 2017) and confine the Sperry Glacier melt season to July, August, and September. Glaciological observations suggest that although May and June can be warm enough to generate melt at Sperry Glacier, the deep snowpack is not necessarily warmed with pore space filled to saturation, and therefore melt does not necessarily run off the glacier."

(now P8.L19) "To test the sensitivity of our results to this assumption, we computed a summer linear regression using MJJAS PDD. Ultimately, the median difference between mass balance values produced by the MJJAS versus JAS regressions was 0.04 m w.e. yr$^{-1}$."

3. Discussion of the mass balance regression and the interpretation of the increase of local topography to the mass balance: The regression model is based on two proportionality factors (ms, mw). The discussion implicitly presumes both factors constant over time, but this needs to be addressed more comprehensively.

This comment is rooted in a misunderstanding, which reflects shortcomings within the paper that we have addressed by making the following three edits:

**(1)** The assignment of time-constant proportionality factors is now more explicit and purposeful:

(now P8.L23) "The linear regression quantifies the 2005-2014 relationship between Sperry Glacier and regional climate, and is fixed with respect to this time. Yet we have hypothesized that this relationship changed as the glacier retreated. By fixing the proportionality factors $m_s$ and $m_w$ to the modern glacier-climate relationship, and then forcing this modern regression with historic climate data, we test our hypothesis that the glacier-climate relationship changed as the glacier retreated."

(now P9.L1) "Attributing the hypothesized change in the glacier-climate relationship to the increasing influence of local topographic effects requires inspection of topographically influenced processes at Sperry Glacier."

- Winter proportionality factors might be variable (Galos et al., 2017; Huss et al., 2008), causing random errors, but systematic errors due to changes in large and meso scale atmospheric flow patterns (Huss et al., 2010) will alter the mass balance regression and weaken the argument of the increasing influence local topography to the mass balance. Can the authors exclude systematic changes of mw?

**(2)** We clarified how systematic changes of winter precipitation are captured by the snow data used as input to the linear regression:

(now P7.L24) "The 1950-2014 period is long enough to encompass meso-scale changes in atmospheric flow patterns, which elsewhere have been shown to have an important impact on winter accumulation and glacier mass balance (e.g. Huss et al., 2010). The peak SWE data we analyzed have been shown to reflect such meso-scale, decadal shifts in snow (McCabe and Dettinger, 2002; Pederson et al., 2011; Selkowitz et al., 2002)."

- Degree-day factors (i.e. the summer proportionality factor) are not constant over time, if the glacier surface area is largely changing. The major reason for this systematic change of the degree-day factor is the albedo feedback (e.g. Naegeli and Huss, 2017). An approximation to this feedback is the change of accumulation area ratio (AAR). Results of the geodetic survey show a thickening of the accumulation area and a concurrent glacier retreat, which means that the accumulation area remained rather constant, while the glacier lost wide parts of its ablation area. Hence, in relation to the total glacier area the AAR increased, resulting in higher mean albedo of the glacier and thus a lower degree-day factor. This effect would indeed strengthen the finding of the increasing influence local topography to the mass balance. (Interestingly, in all studies I'm aware of, the albedo feedback increases melt because the glaciers generally lose their accumulation areas due to rising equilibrium line altitudes.)

(3) We included this valid, strengthening point the Discussion:

(now P11.L30) "Glacier elevation changes reflect both surface mass balance and ice flow processes (Cuffey and Patterson, 2010). However, if we were to attribute geodetic results solely to surface mass balance, then our results (Fig. 3) suggest that the equilibrium line altitude remained relatively constant from 1950-2014 despite climate warming. Thus as Sperry Glacier retreated, its accumulation area ratio (Cogley et al., 2011) increased. With an increased fraction of the glacier remaining snow-covered throughout the melt season, the average glacier albedo increases (Naegeli and Huss, 2017). Such time changes in glacier albedo must have affected the summer proportionality factor, i.e. the amount of area averaged melt relative to regional summer temperature, which in part explains the discrepancy between the linear regression and the 1950-1960 mass balance (Fig.5c)."

**Introduction**

This chapter must more elaborate on the peculiarities and the definition of a small glacier, which in the manuscript seems to be synonymous to a cirque glacier and a very small glacier.

Introduction edited to define and related small glaciers to cirque glaciers:

(now P1.L21): "However, prior studies of small (i.e. <0.5 km2) mountain glaciers, which are often located in cirques…"

Whatever classification is used, the important message in this study is that the local topography has a high influence on the accumulation regime of the Sperry Glacier. In the last paragraph the authors describe briefly the areal change of Sperry Glacier and formulate their research question. At this point I suggest introducing the term accumulation area ratio (AAR) (Cogley et al., 2011) as it (i) presumably describes that the glacier lost its ablation area while the accumulation area almost remained constant and (ii) this fact is crucial to interpret the findings later in the manuscript.

We have added the accumulation area ratio topic to our Discussion, but opt not to include it in the Introduction, so as to keep the opening text streamlined and focused.

**Methods**

I suggest adding a paragraph that the two mass balance methods used (geodetic and glaciological) consider different processes of mass change (e.g. Klug et al., 2018; Zemp et al., 2013). In this study the differences will presumably be smaller than the given errors. The exclusion of methodological differences will support the discussion of the regression model later in the manuscript.

Text speaking to the different processes of mass change represented in the glaciological and geodetic mass balances added:

(now P6.L15): "The results of this calibration, which utilized the 2005-2014 geodetic mass balance calculated in this study to correct the absolute magnitude of annual and summer glaciological balances without losing seasonal/annual variability (Zemp et al., 2013), are reported in Table 1. Such calibration ensures that systematic errors in the glaciological method are rectified and englacial and subglacial mass changes not measured at surface stakes are accounted. Details on this calibration are provided in the Supplement."

In Eq. 1 the authors derive the geodetic mass balance based on the initial glacier area. By convention (Cogley et al., 2011), the mean area between initial and final state is used (Andreassen et al., 2016; Klug et al., 2018; Lang and Patzelt, 1971; Zemp et al., 2013 etc.), thus a recalculation of the geodetic mass balance is required.

Equation and text corrected on (now P4.L11).

We recalculated geodetic mass balances using the initial glacier area ($A_{t1}$) instead of the average glacier area ($A$), and the reported values did not change. An example for the 2005-2014 geodetic mass balance is provided here to illustrate:

$$\Delta B_a = \frac{\Delta V}{A}\left(\frac{\rho_i}{\rho_w}\right) - \frac{\Delta V}{A_{t1}}\left(\frac{\rho_i}{\rho_w}\right)$$

$$\Delta B_a = \frac{900,000\ m^3}{860,000\ m^2}(0.9) - \frac{900,000\ m^3}{830,000\ m^2}(0.9) = 0.03\ \text{m w.e.}$$

This difference, when expressed as an annual average over 2005-2014 (0.003 m yr$^{-1}$), is less than the uncertainty on the geodetic mass balance (0.03 m w.e. yr$^{-1}$) and therefore does not change the geodetic mass balance reported for 2005-2014 in Table 3 (-0.10 $\pm$ 0.03 m w.e. yr$^{-1}$).

**Results**

Section 4.3 supports the albedo feedback mentioned above.

The albedo feedback is now discussed on (now P12.L2-7)

**Figures & Tables**

Figure 1a: Explain HCN.

Done.

Figure 1b: Give exact date of the aerial image. Add the location of the mass balance stakes and the meteorological station.
Done.

Figure 4b: Indicate source of the peak SWE data.
Done.

Figure 6: Add the glaciological mass balance values.
Done.

Figure 10: Add labels a-d.

Figure S2: What is the added value of this Figure? The grey line is hardly visible.
Figure removed.

Figures 5, 6, S2: Figures do not depict continuous data as suggested by the x-axis. Use a bar chart instead of a line graph.
Discrete points on the line graphs convey that these are annual data, and we have added text to the new Figure 5 caption to clarify this point. (The new Figure 5 combines the original Figure 5 and Figure 6, and the original Figure S2 has now been omitted.)

Table 1: Add columns of ELA and AAR.
ELA does not follow an elevation band on Sperry Glacier. See snowline traced in Figure 1b. Thus we decided that listing the seasonal snowline (i.e. annual ELA) as a single elevation in this table would not be meaningful.

We added AAR reported by Clark et al. (2017) to Table 1.

Table 2 and S2: As glacier hypsometry is generally not normally distributed the median elevation is preferable to the mean elevation.
Median results are now reported in Table 2 and Table S1.

Table S3: Explain mw in the table capture.
Done.

**Specific comments**
P 1, L 15 and elsewhere in the manuscript: Are negative mass loss rates a mass gain? This is pedantic but I suggest using neutral formulations that do not compete with the sign of the corresponding value.
Fixed.

Glaciologists traditionally use m w.e. as mass balance unit. Let's convert to kg/m² (≡ mm w.e.), because this is the SI unit and outside the glaciological community nobody understands w.e.

We acknowledge this point, but opt to stick with the glaciological convention of m w.e. to make our work directly comparable with other glacier mass balance studies (Andreassen et al., 2016; Huss et al., 2009, 2010; Klug et al., 2018; Zemp et al., 2013).

P 2, L 20: relative to what?

Text edited from (now P2.L29): "We leverage relatively unique observations…" to "To test this hypothesis, we leverage field measured (glaciological) surface mass balance record and repeat geodetic mass balances."

P 6, L 26: The air temperature lapse rate must be a negative value.

Fixed.

P 6, L 27: Rephrase the sentence. 7 years ago is not recently.

Removed the words "recently installed."

P 7, L 1: Was the glacier surface on average 35 m lower in elevation or just the terminus?

Changed to (now P7.L13) "…the surface of Sperry Glacier was on average 35 m lower in elevation…"

P 7, L 4: Change the units to °C and maybe rephrase the last part to "…, and cumulatively -24°C and -14°C changes to PDD."

Done.

P 9, L 5+6: Replace million and billion by 106 and 109, respectively.

Done.

P 9, L 9: Link to Table 3?

Fixed.

P 9, L 28: Rephrase the sentence beginning with "Enough years are positive…"

Rephrased to (now P10.L28) "Sufficient individual years are positive during the mid-20[th] century to yield positive averages…"

P 10, L 4: Explain how you derived this number.

Radiation calculation method explained in section 3.6 Shading, avalanching, and wind-drifting (now P9.L1-9).

P 11, L 1: Explain the meaning of uniform mass balance gradients. Gradients in the ablation area are usually different from those in the accumulation area, mainly because of the higher albedo in the latter (Kaser et al., 1996; Kuhn et al., 1999).

Phrase removed as it did not accommodate the complexity you note, nor did it advance the purpose of the paragraph, which is to introduce the mass balance gradient at Sperry Glacier.

Table 2 and S2: As glacier hypsometry is generally not normally distributed the median elevation is preferable to the mean elevation.
Median results are now reported in Table 2 and Table S1.

Andreassen, L. M., Elvehøy, H., Kjøllmoen, B., and Engeset, R. V. (2016). Reanalysis of long-term series of glaciological and geodetic mass balance for 10 Norwegian glaciers. *Cryosphere* 10, 535–552. doi:10.5194/tc-10-535-2016.

Cogley, J. G., Hock, R., Rassmussen, L. A., Arendt, A. A., Bauder, A., Braithwaite, R. J., et al. (2011). *Glossary of mass balance and related terms*. Paris: IHP-VII Technical Documents in Hydrology No. 86, IACS Contribution No. 2, UNESCO-IHP.

Cuffey, K. M., and Patterson, W. S. B. (2010). *The Physics of Glaciers*. 4th ed. Oxford: Elsevier.

Huss, M., Bauder, A., Funk, M., and Hock, R. (2009). Determination of the seasonal mass balance of four Alpine glaciers since 1865. *Mitteilungen der Versuchsanstalt fur Wasserbau, Hydrol. und Glaziologie an der Eidgenoss. Tech. Hochschule Zurich* 113, 11–29. doi:10.1029/2007JF000803.

Huss, M., Hock, R., Bauder, A., and Funk, M. (2010). 100-year mass changes in the Swiss Alps linked to the Atlantic Multidecadal Oscillation. *Geophys. Res. Lett.* 37, 1–5. doi:10.1029/2010GL042616.

Klug, C., Bollmann, E., Galos, S. P., Nicholson, L., Prinz, R., Rieg, L., et al. (2018). Geodetic reanalysis of annual glaciological mass balances (2001-2011) of Hintereisferner, Austria. *Cryosphere* 12, 833–849. doi:10.5194/tc-12-833-2018.

McCabe, G. J., and Dettinger, M. D. (2002). Primary Modes and Predictability of Year-to-Year Snowpack Variations in the Western United States from Teleconnections with Pacific Ocean Climate. *J. Hydrometeorol.* 3, 13–25. doi:10.1175/1525-7541(2002)003<0013:PMAPOY>2.0.CO;2.

Naegeli, K., and Huss, M. (2017). Sensitivity of mountain glacier mass balance to changes in bare-ice albedo. *Ann. Glaciol.* 58, 119–129. doi:10.1017/aog.2017.25.

Pederson, G. T., Gray, S. T., Ault, T., Marsh, W., Fagre, D. B., Bunn, A. G., et al. (2011). Climatic controls on the snowmelt hydrology of the northern Rocky Mountains. *J. Clim.* 24, 1666–1687. doi:10.1175/2010JCLI3729.1.

Selkowitz, D. J., Fagre, D. B., and Reardon, B. A. (2002). Interannual variations in snowpack in the Crown of the Continent Ecosystem. *Hydrol. Process.* 16, 3651–3665. doi:10.1002/hyp.1234.

Zemp, M., Thibert, E., Huss, M., Stumm, D., Rolstad Denby, C., Nuth, C., et al. (2013). Reanalysing glacier mass balance measurement series. *Cryosphere* 7, 1227–1245. doi:10.5194/tc-7-1227-2013.

---

## Author Comment (AC2) · 18 May 2018

Author responses in blue.

Review of Florentine et al. - Local topography increasingly influences the mass balance
of a retreating cirque glacier (09/04/2018)

Overview ——— The authors present an analysis of the mass balance of a small
mountain glacier in North America with respect to regional climatic trends and the
influences of local topography. Using a combination of geodetic and glaciological approaches,
the authors observe a small reduction of average mass loss in the glacier
between three periods of time, namely 1950-1960, 1960-2005 and 2005-2014, despite
being strongly distinct to regional mass balance estimates, based upon positive degree
days and winter precipitation records since 1950. The paper is concise and very well
written with a good amount of thought given to potential sources of error and uncertainty.
The topic is relevant to the cryospheric community given the decreasing size
of many mountain glaciers and the increased role of local factors on the glacier-wide
mass balance. I believe the manuscript to be of sufficient quality and relevance for
publication in The Cryosphere and think that the work should be published.

Nevertheless, I believe the authors need to address more the role of the local affects
in promoting the slow down of mass loss for a glacier such as Sperry Glacier. The
manuscript does well to highlight the assumptions of regional climate and the stark
contrasts with the local mass balance, but needs to further highlight the role of local
events and their spatial domain, so that such processes may be more readily compared
with the observations of mass balance across the glacier. The authors present a nice
case study of the glacier with a likely increase in the contribution from local factors,
though I believe the information on local factors needs to be quantified more and presented
more clearly for the reader on maps that can be interpreted. Various figures do
not contribute enough the arguments of the manuscript and thus limits the conclusions
that are drawn to some extent. The results presented on regional mass balance estimates
vs. the geodetic changes imply that, although local topography/meteorological
conditions is likely having a greater influence on mass balance, Sperry Glacier has,
since 1950 at least, been decoupled from the regional climate.

One of the largest drawbacks of the paper is the lack of introduction and, importantly, discussion of the literature to contextualise the findings in the Glacier National Park. For a revision of this manuscript, I believe more needs to be added to compare Sperry Glacier with regional vs. local influences as published elsewhere in the world. I provide some suggestions of relevant research that could be included here and specific questions and comments that should be addressed before publication in the journal. I consider, on the whole, a minor revision, though would like to see some larger changes to the introduction, figures and discussion of the local effects on the glacier.

Thank you for the insightful comments and pointers to relevant literature. We have made many of the changes you recommended, namely to bolster the Introduction/Discussion and to revise Figures.

These revisions provide improved, more explicit explanation of the spatial domain and particularities of local topographic processes (avalanching, wind drifting, and shading) at Sperry Glacier. The revisions also provide stronger linkages to glacier mass balance studies elsewhere on Earth.

Abstract ——— P1.L17 ". . .closely predicts the geodetically measured mass loss from 2005-2014." Please quantify 'close'.
P1.L14 Edited to: "…closely (i.e. within 0.08 m w.e. yr$^{-1}$) predicts…"

P1.L18-20 Overestimates 1950-60 Mass balance – does this not imply that recent mass balance (2005-2014) can be better explained by regional climate? See later comment regarding interpretation of the results with respect to local vs. regional influences.
We write explicitly that recent mass balance (2005-2014) can be successfully explained by regional climate:

P1.L13 "A correlation of field-measured mass balance and regional climate variables closely (i.e. within 0.08 m w.e. yr$^{-1}$) predicts the geodetically measured mass loss from 2005-2014."

The overestimation implies that the way regional climate relates to recent mass balance (2005-2014) is different than the way regional climate relates to historic mass balance (1950-1960).

We unpack the meaning of this overestimation in the Discussion, using additional analyses. Since the overestimation itself does not implicate the increasing influence of local effects, we edited the word "This" on P1.L16, which referred to the liner regression overestimation, to "Our analysis," which refers to the analysis presented in the Discussion (please see the new, revised Figure 7).

Intro —— P1.L23 Yes, radiation inputs are dominant in summer for most glaciers, but I think it would be appropriate to rephrase this as radiation and air temperature driven (this can often be the case for coastal environments/ maritime glaciers where longwave and turbulent fluxes can sometimes dominate) – Also because you relate mass balance to PDDs in this paper (though the PDDs and shortwave radiation are, of course, well related).

P1.L20 Edited to "…ice mass losses are ultimately controlled by radiation *and air temperature during summer.*"

P1.L27 What about Carturan et al. (2012) and the case of Italy's lowest elevation glacier, Montasio Occidentale? I think this should be included in your introduction. Also, what about the variability in meteorological conditions and energy balance processes for a small glacier such as shown by Hannah et al. (2000)? Some mention to the changing role of glacier energy balance would be good.
Sentences referring to this literature added:

P1.L25 "Analysis from Montasio Occidentale Glacier in the Italian Julian Alps demonstrated that avalanche-fed, shaded glaciers can exist at low elevations otherwise climatically unsuitable for the persistence of glacier ice (Carturan et al., 2013). Analysis from a network of on-ice automatic weather stations on a cirque glacier in the French Pyrenees concluded that topographic effects may exert more control on surface energy budgets - and thereby melt - than regional lapse rates in air temperature and moisture (Hannah et al., 2000)."

P2.L2-3 Hoffman 2007 found no trend with winter precipitation in the 20th Century. I think you should add more details about the glacier sizes during that study. Was there a decline in winter precipitation for their study? How else can you state the role of local processes in your following sentence?
P2.L2-3 "In Colorado, glacier mass balance at very small (<0.2 $km^2$) glaciers showed no statistically significant correlation to winter precipitation during the $20^{th}$ century, suggesting that winter mass inputs were not connected to regional winter precipitation in a straightforward, linear manner (Hoffman et al., 2007). This result instead implied the importance of processes driven by local topography…"
In your manuscript, you are talking about increased sensitivity to regional climate for your earlier periods of observation (during 20th Century). How will Hoffman's findings compare to your results? I think this is something to be discussed later in your work.
We related Hoffman's findings to the Discussion (see now P13.L18)

P2.L8-10 I think a reference here would be suitable
P2.L8-10 (now P2.L11) Reference Haugen et al. (2010) added.

P2.L10-14 Again some more reflection on Hoffman's findings would be good for this introduction. Is Andrews Glacier facing south?
P2.L10-14 (now P2.L12) "Andrews Glacier, an east-facing <0.2 $km^2$ glacier in Colorado…"
Are the radiation loads or some other factor sufficient to explain why regional climate still outweighs the local influences? In addition, what about the controls of the atmospheric boundary layer adjustment for the retreat of a mountain glacier? For example, one could expect a diminishing katabatic boundary layer for fragmenting glaciers (see Carturan et al. 2015 - TC) and an increase in the sensitivity of the near-surface air to air temperature fluctuations outside the boundary layer (see Greuell and Böhm, 1998 and Shaw et al., 2017 – JoG).
Sentences added to reflect on the meaning of Hoffman's findings on Andrew's glacier:

P2.L10-14 (now P2.L15) "Mechanistically, this could be because Andrews Glacier is small enough to have a diminished katabatic boundary layer at its surface, leaving the glacier sensitive to ambient summer temperatures (Carturan et al., 2015). Regardless, this example shows that regional climate can remain the primary driver of net mass balance, even when local topography plays a strong role."

Study site ———– P2.L24 Please give coordinates centred on the glacier
P2.L24 (now P2.L27) Coordinates added.

P3.L3 Historical photographs from which year?
P3.L3 (now P3.L6) Edited to "Historical photographs and analysis from 1914 (Alden, 1914)…"

Methods ——— P3.L27 Can you provide more information about the area size and locations on 'stable' ground that was used for the co-registration. Was there presence of landslips/rockfalls that may account for some of the large vertical differences? Consider including a figure to the supplementary information.
P3.L27 (now P3.L31) Figure added to the supplement. Text edited to "over stable (i.e. not prone to landslide/rockfall)…bedrock (Fig.S1, Supplement)."

P4.L3 What aerial imagery?
P4.L3 (now P4.L7) Edited to "Worldview aerial imagery (Fig. 1b)…" now shown in Figure 1b. We had mistakenly omitted this image source.

P4.L11 Can you elaborate here why you decided to apply the Huss Approximation for glacier ice density and not use the values provide from the glacier itself in the case of Clark et al., 2017 with an error range?
P4.L11 (now P4.L15) The Clark et al. (2017) density was that reported by Cuffey and Patterson (2010), and was not from the glacier itself. We deleted this sentence as it added little value and confused.

P4.L25 How does the quantity of stable bedrock, to provide more certainty in elevation differences, vary with elevation in your study site?
P4.L25 (now P4.L23) "The quantity of stable bedrock points varied from 14,472 in the lowest elevation band to 1,825 points in the highest elevation band."

P5.L23 Can you also give a bit more detail about how this information was filled? You mean to say that (as with lines 26-27) that you used the 2014 generated DEM elevations for these areas that were missing? You state that the rate of mass change (1950-2014) is the same. . . but you mean to say there is no mass change, as you use the same DEM values? Or there is no DEM information for these years, but just a rate of change for 2005-2014 applied? Perhaps I am missing something clear with this paragraph, though perhaps you can state it more directly?

P5.L23 (now P5.L27) Edited to "By using modern elevation change rate results to infill missing historic data…"

Also, including the elevations that were missing from the 1950-60 photographs and a map of the missing areas with the supplementary information would be valuable for interpretation of your results, despite that you state the differences are small and within the error margins.

The missing upper section is shown in Figure 3 glacier maps (Fig.3a and Fig.3b).

P6.L10 Please provide further details on the calibration of the biases in raw mass balance data on Sperry Glacier here. Please plot the mass balance measurements (I assume stake data?) in the map of Figure 1b.

Stake locations added to Figure 1b. We added text and a supplementary methods section to provide further calibration details:

P6.L10 (now P6.L15) "…which utilized the 2005-2014 geodetic mass balance calculated in this study to correct the absolute magnitude of annual and summer glaciological balances without losing seasonal/annual variability (Zemp et al., 2013), are reported in Table 1. Such calibration ensures that systematic errors in the glaciological method are rectified and englacial and subglacial mass changes not measured at surface stakes are accounted. Details on this calibration are provided in the Supplement."

We also caught an error in our reporting, which led to corrections in Table 1.

P6.L15-16 A very minor point, but can SWE be considered a meteorological variable ('meteorological data')?

Astute point. Manuscript edited throughout to ensure SWE is referred to as climate data rather than meteorological data.

P6.L20-22 I do not argue with the use of Kalispell to represent PDDs for this analysis, though are there no other meteorological information within the greater area that may aid construction of ensemble PDDs for 'regional' climate? Is there any indication of local processes at Kalispell that may obscure the relationships you derive? Furthermore, are there any off-glacier meteorological observations in the basin of the study site that give indication to the representation of PDDs derived from Kalispell?

We have no indication that local processes at Kalispell would disrupt temperature profoundly or systematically enough to obscure the relationship we derived. In fact, our analysis showed that the similarity between lapse-corrected Kalispell temperatures and in-situ measurements from the Sperry Glacier met station was statistically significant:

P6.L20-22 (now P7.L4) "Comparison of lapse-rate-corrected temperatures derived from the Kalispell meteorological station with the 7 year record of in situ July, August, September temperatures measured at the Sperry Glacier meteorological station (Fig. 1b) yielded no statistically significant difference at the 99% confidence interval (p<0.01), and the distribution of residuals was normal."

Still, you make a fair point. Assuming that the Kalispell model is representative of regional climate conditions is a source of uncertainty, which we account for with the linear regression confidence intervals:

P6.L20-22 (now P8.L15) "…we used the upper and lower confidence intervals to compute maximum and minimum possible mass balances….This accounting accommodated uncertainty due to (a) the discrepancy between snow and meteorological station locations and Sperry Glacier…"

We also added a figure to the Supplement and the following sentence to the text, which support our treatment of Kalispell-derived PDDs as definitive of regional conditions:

(now P6.L30) "Temperatures measured at Kalispell are highly representative of regional climate as reflected by gridded values from North American Regional Reanalysis output (Supplement, Fig. S3a)."

P6.L23 Partly related to the previous point, how did you 'vet' the Kalispell data record using the Sperry Glacier one? The application of an above-ice AWS would reveal not only a reduced average temperature compared to that outside the thermal influence of the glacier, but also a 'dampened' diurnal cycle (again see Carturan et al., 2015- TC).
We clarified our meaning:

P6.L23 (now P6.L29) "To assess the representativeness of the Kalispell temperature record relative to the glacier site, we compared this shorter, discontinuous record with the longer, continuous Kalispell record."

The glacier site AWS station is off-ice, which is now shown on Figure 1b.

P6.L26 Did you utilise only JJA to correct the temperature using the vertical temperature lapse rate? See above point. Where was the Sperry AWS? Plot also in Figure 1b.
Yes, we utilized JAS data to solve for the vertical lapse rate used to correct JAS temperatures. Sperry met station location added to Figure 1b.

P7.L11 State which observations specifically were used for this analysis.
P7.L11 (now P7.L28) "Based on an analysis of the historical snow data from the region against observations from Sperry Glacier, we chose to use Mount Allen snow data for our mass balance regression."

P8.L3 Is there evidence from the glaciological observations on Sperry Glacier to suggest that the melt season does not (in a 'typical year') begin earlier than July?
We now cite glaciological observations to justify our definition of melt season months as July, August, and September and report how JAS results.

P8.L3 (now P7.L8) "The Kalispell record reports continuous daily average air temperatures back to 1950 for the melt season summer months. We follow the convention defined by previous GNP melt modelling (Clark et al., 2017) and confine the Sperry Glacier melt season to July,

August, and September. Glaciological observations suggest that although May and June can be warm enough to generate melt at Sperry Glacier, the deep snowpack is not necessarily warmed with pore space filled to saturation, and therefore melt does not necessarily run off the glacier."

P8.L10-12 I think the information on your assessment of the local processes (i.e. avalanching and snow drifts) are lacking in this section. Again, information about where observations from Clark et al. (2017) are in relation to the qualitative information used would be highly informative for the reader to comprehend the arguments about the increased role of local conditions/processes in governing the mass balance of Sperry Glacier.
We have improved illustration of local processes in now Figure 6.
Results ——— P9.L2-3 Is there any information on the development of debris-covered ice at the terminus of the glacier, which may enhance or diminish local melt rates?
P9.L2-2 (now P9.L23) "No development of debris cover at the glacier terminus that might explain this decreased thinning rate is evident."

P9.L26-27 Perhaps I miss something here, but 1950-60 and 2005-14 don't seem all that comparable in Table 3. Admittedly there are not huge differences. . . but still statistically significantly different, no?
Clarified our meaning:

P9.L26-27 (now P10.L25) "…average mass change rates at Sperry Glacier for 2005-2014 (-0.10 $\pm$ 0.03 m w.e. yr$^{-1}$) and 1950-1960 (-0.22 $\pm$ 0.12 m w.e. yr$^{-1}$) were comparable, i.e. within error bounds."

P10.L4-7 This is a nice evaluation of potential energy losses. Could you provide the values in Wm-2 as would be more typically reported for studies for glacier energy balance? Is the 0.36 m w.e. deficit for the whole glacier over the whole period?
Converted and clarified:

P10.L4-7 (now P11.L4) "…decreased by 118,605 kJ m$^{-2}$ (approximately 15 W m$^{-2}$)…"

Added phrase to clarify that the deficit was for the whole glacier over the whole period:

P10.L4-7 (now P11.L6) "…this energy deficit translates to 0.36 m w.e. m$^{-2}$ less potential melt for the summer melt season averaged over the entire glacier…"

P10.L10-13 Can you indicate what evidence you have for wind effects on glacier mass balance? Evidence from Figure 8 show a potentially important source of local mass input, though your reported results here don't strongly support the information you have already presented. Figure 8 does very little alone to bring together your ideas as it does not contain information relating to the elevation ranges of the wind drift snow and thus,

the values of wind scour/accumulation effects in the following sentence suggest large spatial variations in mass balance, but do little more. Perhaps you could provide data on typical wind direction for the basin, based upon the Sperry Glacier AWS records. Do these tie in with more recent wind deposits apart from what was seen in the historic images? Is there a pattern here that might explain the mass balance trends which don't conform to a regional mass balance assumption? I think compiling a more useful figure with combinations from Figure 7 and 8 with a digitised map could be appropriate.

P10.L10-13 (now Figure 6) The new Figure 6 is annotated with more information about wind in the glacier basin.

Furthermore, could you provide information on the exposure of the Sperry Glacier by topographic information from your DEM? For example, the 'Sx' parameter following Winstral et al. (2002) could be appropriate for looking at the potential effects of wind on the initial deposition of snow in the winter mass balance and the re-distribution in summer. Is there anything to suggest these effects have changed through the decades, or just the reduction of glacier ice at lower elevations slowing the total glacier mass loss? A full-scale analysis would be beyond the scope of the paper, though I believe more is required here to argue to the case of local vs regional effects.

The lack of quantitative evidence for mass added to the glacier by wind drifting snow and avalanching indeed limits our argument. However, our task was to assess the time evolution of local influences in bulk. Quantifying and partitioning that evolution would be a logical next step, which we now outline in the Discussion:

(now P13.L7) "Here we have examined the time evolution of local effects in bulk. To quantitatively partition the mass impact of discrete processes, future work could assess the evolution in geographic parameters for wind (e.g. Winstral et al., 2002) and avalanche (e.g. Carturan et al., 2013) effects."

P10.L17-18 Again, this gradient really argues the case for avalanching of material, though information about the location of the observations would be both useful and interesting.

P10.L17-18 (now P11.L9-15) These results now refer to the new Figure 6, which conveys the location of avalanche observations.

Discussion ———— P10.L30 I think it could be argued, particularly based upon your results of Figure 6, that even since 1950, the small size and topographical characteristics of Sperry Glacier have limited its mass loss. A 'small glacier' is relative to whom you ask, though in my experience, 3.24km2 (the size of the glacier in the historical images) is small. The key aspect of this paper suggests that further retreat has increased the role of local topography. The evidence from geodetic change, which has been well assessed with regards to errors, would suggest that, indeed, the glacier mass loss has slowed despite conditions favouring its demise. Nevertheless, a point to make here is that, compared to what the regional trends in PDDs and winter accumulation would suggest, the glacier-wide mass balance of Sperry Glacier has since 1950 (and likely

further back in time) been somewhat decoupled from what the regional climate would prescribe. One may have to assess a much more historic form of the glacier, with much greater size, to identify a stronger relationship to regional climatic trends.

We agree, and edited a sentence that summarized this point so that it is not lost on the reader. Rather than conflating the fact that Sperry Glacier mass balance has been affected by local processes since 1950 with the fact that winter accumulation seems to be the more influential local process (that point is now separately discussed in the next paragraph), now the sentence simply reads:

P10.L30 (now P12.L28) "These results demonstrate that local mass balance processes have apparently played a strong role at Sperry Glacier since 1950, and that role strengthened as the glacier retreated."

P11.L8 Why is there a sudden steepening of the summer mass balance for the top of the glacier? Is there not just a lower mass balance record between the 2420 and 2550 values in Figure 9? What can explain this?

P11.L8 (now P11.L16) "Field-measured summer data at high elevations are sparse, but these summer point balances also show a sudden steepening of the summer mass balance gradient above 2475 m (Fig. 7c), likely because these high reaches of the glacier are shaded by the headwall."

P11.L10-17 I think I miss something with regards to the representation of information in this figure (10). The regional 'lapse rates' (a term I would consider changing to mass balance gradient) is that derived from Figure 5, I presume? However, for which periods of time are these mass balances shown, the complete period? Maybe the reflection of mass balance relating to glacier hypsometry needs some further clarification here.

Figures combined and edited to clarify. The revised Figure 7 is now annotated to describe how we applied field-measured local and regional mass balance gradients to 1950 and 2014 glacier hypsometry to parse local from regional mass balance effects.

P11.L19 79% of what, exactly? Is this evidence shown in Figure 10? To me, these contributions of summer and winter do not show such a strong difference as 79:21.. Again, please provide greater clarification to the reader.

Text and now Figure 7 edited to clarify:

P11.L19 (now P12.L31) "The impact of topographic effects is not evenly partitioned between seasonal components. The differences between the solid and dotted lines in Figure 7b and Figure 7d illustrate local effects: suppression of summer melt (light red area), surplus in winter accumulation (light blue area), and net mass balance enhancement (light gray area). We find that winter local effects (i.e. the surplus in winter accumulation due to avalanching and wind loading) account for 79% of the discrepancy between the mass balance defined by regional mass balance gradients alone versus that defined by both regional and local gradients. Summer local effects (i.e. the mediation in summer melt due to shading) accounted for 21% of the discrepancy."

P11.L26 The findings of Mattias Huss are highly relevant in this discussion, though I would like to see more of this discussion section related to the information which is in (and should be added to) the introduction. See comments on introduction section for suggestions on linking your discussion more with past work on small mountain glaciers. Discussion bolstered accordingly, with literature presented in the Introduction added. P11.L26 (now P13.L10) "Sperry Glacier's increasing sensitivity to local mass balance drivers is consistent with studies of 20[th] century glacier change elsewhere in the Rocky Mountains (DeBeer and Sharp, 2009), and is commensurate with modeled projections of future changes to cirque glaciers in the Swiss Alps."

P13.L32 (now P13.L17) "…suggests that local mass balance drivers do not interrupt the synchronicity of glacier response to climate change on global, century-long scales. Indeed, very small (<0.2 km$^2$) glaciers in Colorado showed a strong annual mass balance response to 20[th] century summer temperatures despite being heavily influenced by winter topographic effects (Hoffman et al., 2007). Thus, the evolving relationship between climate and mass balance demonstrated by Sperry Glacier reveals the complexity in interpreting glacier changes in Glacier National Park and at small cirque glaciers elsewhere on Earth, but does not preclude the reality of a climate that is trending toward conditions that mandate glacier disappearance.

Figures ——– Generally, the figures are well presented and clear. I have a few suggestions about combining figures and using others to a greater effect:

Figure 1b should display the locations of the mass balance observations for reference of the reader. Figure 1b: Done.

Figure 2 could be combined with Figure 1 and set as a larger typeset figure for the article. The map information in both is relevant to the location of things in your study basin which can be referred to many times to aid your conclusions. Figure 2: We chose to leave Figure 2 as a stand-alone figure, so as to maintain the non-square aspect ratio (width) and thereby include moraines on the edge of the photo.

Figure 3: Can you change the elevation interval averages and error bars (light green) to a different colour? Perhaps red? Figure 3: Green changed to red.

Figure 4: I think there is perhaps a bit too much information contained within the caption which is sufficiently explained in the text. Figure 4: Caption edited for brevity.

I think it could be useful to combine Figures 5 and 6 and stack them vertically as they

show similar information and time-series trends. The results of Figure 6 are clearly very strongly influenced by the winter precipitation in Figure 5 and combining these figures would aid interpretability.

Figure 5: Combined with Figure 6. We agree that this will aid reader interpretation.

As mentioned in the review, Figure 7 and Figure 8 provide some potentially useful information, though it is not used to great effect. Some information about the locations, elevations and extent of these observations (where possible) would be useful in comparison to the digitised glacier map of the same (or approx.) time period. This should be leveraged to explain some of the mass balance behaviour of Sperry Glacier in your study. As suggested previously, some information about the consistency of wind drifts and/or wind scouring (and a better map of the location(s) could be useful). For example, how could we then treat these effects in future modelling efforts to better represent the future of small glaciers? Should we/could we at all?

Figure 7/8 (now Figure 6): Figures combined as recommended. Location and elevation information added.

Figure 9: I think this figure is very important to the suggestion of localised processes, particularly during winter. However, it is not clear to me, the derivation of regional and local 'lapse rates' here. Again, I would suggest an alternative term for this too, perhaps mass balance gradient.

Figure 9 (now Figure 7): Language changed as advised.

Figure 10: Again, I think this a good and valuable figure to the paper. I'm not convinced that the text explains the information of the figure fully, or perhaps I miss something. Please try and make this clearer to the reader. For example, which are the gradients (here referred to as gradients) which are mapped onto the glacier hypsometry in these two years?

Figure 10 (now Figure 7): Figure annotated and text added to clarify.

Supplementary information ——————————- I'm not completely sure what Figure S2 is showing to support your work here and appears to only show a corrected result despite being intended to show a pre- and post-correction.

Figure S2: Agreed; the small change between pre- and post-correction is sufficiently communicated in the test. This supplement figure served no clear purpose and so has been removed.